# Supported Policy Optimization for Offline Reinforcement Learning

**Jialong Wu**[1], **Haixu Wu**[1], **Zihan Qiu**[2], **Jianmin Wang**[1], **Mingsheng Long**[1][✉]
[1]School of Software, BNRist, Tsinghua University, China
[2]Institute for Interdisciplinary Information Sciences, Tsinghua University, China
{wujialong0229,qzh11628}@gmail.com, whx20@mails.tsinghua.edu.cn
{jimwang,mingsheng}@tsinghua.edu.cn

## Abstract

Policy constraint methods to offline reinforcement learning (RL) typically utilize parameterization or regularization that constrains the policy to perform actions within the support set of the behavior policy. The elaborative designs of parameterization methods usually intrude into the policy networks, which may bring extra inference cost and cannot take full advantage of well-established online methods. Regularization methods reduce the divergence between the learned policy and the behavior policy, which may mismatch the inherent density-based definition of support set thereby failing to avoid the out-of-distribution actions effectively. This paper presents **S**upported **P**olicy **Op**T**imization (SPOT), which is directly derived from the theoretical formalization of the density-based support constraint. SPOT adopts a VAE-based density estimator to explicitly model the support set of behavior policy and presents a simple but effective density-based regularization term, which can be plugged non-intrusively into off-the-shelf off-policy RL algorithms. SPOT achieves the state-of-the-art performance on standard benchmarks for offline RL. Benefiting from the pluggable design, offline pretrained models from SPOT can also be applied to perform online fine-tuning seamlessly.

## 1 Introduction

Offline RL [28, 29], where the agent learns from a fixed dataset, collected by arbitrary process, not only provides a bridge between RL and the data-driven paradigm but also eliminates the need to interact with the live environment, which is always expensive or risky in practical scenarios [16, 31, 18]. Unfortunately, the absence of environment interaction also raises a number of challenges. Previous work has shown that the extrapolation error of the Q-function queried by out-of-distribution actions significantly degrades the performance of off-policy algorithms [9].

Avoiding out-of-distribution actions, namely to constrain the learned policy to perform actions within the support set of the behavior policy, is essential to mitigate extrapolation error. To meet this support constraint, policy constraint methods [29] to offline RL utilize either *parameterization* [9, 50, 11] or *regularization* [15, 24, 47] techniques. However, there are still several drawbacks in previous methods of policy constraint, limiting their performance and applications. Firstly, parameterization methods involve elaborate designs of the policy parameterization, typically coupled to generative models of the behavior policy, to *directly* constrain actions taken by the learned policy. But these designs intrude into the architecture of policy networks, which may bring extra inference cost and supplementary difficulties to implement and tune offline RL algorithms. Furthermore, as criticized by Fujimoto and Gu [7], these *intrusive* designs complicate causal attribution of performance gains and transfer of techniques between offline RL algorithms or from well-established online RL algorithms. In contrast, for the second category, regularization methods are designed with a non-intrusive or *pluggable*

manner, which is done by simply augmenting a penalty to the actor loss that measures the divergence of the learned policy from the behavior policy [15, 24, 47, 22]. However, these divergence-based regularization methods may mismatch the inherent density-based definition of support set, which applies support constraint *indirectly*. As we show in experiments (Section 5.1), the divergence-based design prevents regularization methods from effectively avoiding out-of-distribution actions and thus limits their performance.

In this work, we aim to design a *pluggable* offline RL method that also *directly* meets the standard formalization of the support constraint based on the density of the behavior policy. We introduce **S**upported **P**olicy **O**p**T**imization (SPOT), a regularization method which can be plugged non-intrusively into off-the-shelf off-policy RL algorithms. SPOT involves a regularization term from the new perspective of explicit estimation of the behavior density. Concretely, SPOT adopts a VAE-based density estimator [19, 37] to explicitly model the support set of behavior policy and presents a simple yet effective regularization term directly applied to the estimated density.

Our method benefits from a closer connection between theory and algorithm, thereby achieving superior performance compared to previous methods. It also profits from the pluggable algorithmic design, which leads to efficient inference and minor algorithmic modifications on top of standard off-policy algorithms. Moreover, a minimal gap between offline learning objectives and standard online learning objectives also enables SPOT to take full advantage of existing online RL algorithms and attain strong online fine-tuning performance after offline RL, exceeding state-of-the-art methods.

The main contributions of this work are three-fold:

- We derive a regularization term for constrained policy optimization in offline RL, based on the theoretical formalization of the density-based support constraint, which directly regularizes the behavior density of actions taken by the learned policy.
- We propose Supported Policy OpTimization (SPOT), a practical algorithm with a neural VAE-based density estimator to implement the proposed regularization term.
- Compared to several strong baselines, SPOT achieves state-of-the-art results on standard offline RL benchmarks [6] and also outperforms previous methods when online fine-tuned after offline RL initialization.

## 2   Related Work

Our work belongs to the family of policy constraint methods in offline RL, where parameterization or regularization is typically utilized. Comparison between previous policy constraint methods and ours has been summarized in Table 1. In addition to the policy constraint methods, we also review behavior policy modeling in offline RL, which is commonly necessary to both parameterization and regularization methods. Lastly, we briefly discuss a broader range of competitive offline RL approaches.

Table 1: Comparison among policy constraint methods.

| Method | Support Constraint | Implementation |
|---|---|---|
| BCQ [9] PLAS [50] EMaQ [11] | Explicit Density Constrained | Intrusive Parameterization |
| BEAR [24] BRAC-p [47] TD3+BC [7] | Implicit Density Constrained | Pluggable Regularization |
| **SPOT (Ours)** | Explicit Density Constrained | Pluggable Regularization |

**Policy constraint via parameterization.** Careful parameterization of the learned policy can naturally satisfy the support constraint. For example, BCQ [9] learns a generative model of the behavior policy and trains the actor to perturb randomly generated actions. The policy parameterized by BCQ is to greedily select the one maximizing Q function among a large number of perturbed sampled actions. EMaQ [11] simplifies BCQ by discarding the perturbation models. PLAS [50] learns a policy in the latent space of the generative model and parameterizes the policy using the decoder of the generative model on top of the latent policy.

**Policy constraint via regularization.** An alternative approach is to use a divergence penalty in order to compel the learned policy to stay close to the behavior policy, such as KL-divergence [15, 47], maximum mean discrepancy (MMD) [24], Fisher divergence [22] and Wasserstein distance [47]. Using divergence penalties alleviates the hard constraint of explicit parameterization but may bear the risk of out-of-distribution actions. Although BEAR [24] attempts to constrain the policy to the

support of the behavior policy, it heavily relies on the empirically-found approximate property of low-sampled MMD. TD3+BC [7] simply adds a behavior cloning (BC) term to the policy update and presents competitive performance on simple locomotion tasks. Note that recent SBAC [48] proposes a new policy learning objective based on performance difference lemma [17], along with a density-based regularization term similar to ours, but our work simply plugs the term into the standard policy training objective to enjoy minimal algorithmic modifications.

**Behavior policy modeling in offline RL.** Most policy constraint methods need to fit an accurate generative model of the behavior policy, to sample in-distribution actions or estimate behavior density. Conditional variational auto-encoders (CVAE) [19, 40] are typically used by past works [9, 50] to sample actions, while EMaQ [11] opts for using an autoregressive model [10] which enables more accurate sampling. On the other hand, policy class of Gaussian [24, 47] or Gaussian mixture models [22] are commonly used to fit and estimate the density of behavior policy. Instead of explicitly fitting the behavior policy, recent approaches have utilized implicit constraint without ever querying the values of any out-of-sample actions [49, 23]. Although MBS-QL [30] uses ELBO to estimate marginal state distribution, to the best of our knowledge, we are unique in estimating the density of behavior policy based on VAE, for its flexibility to capture almost arbitrary class of distributions [20].

**Broader range of offline RL approaches.** Besides policy constraint methods based on parameterization or regularization, there exist more types of competitive offline RL approaches. IQL [23] designs a multi-step dynamic programming procedure based on expectile regression, which completely avoids any queries to values of out-of-sample actions. Pessimistic value methods, such as CQL [26], produce a lower bound on the value of the current policy to effectively alleviate overestimation, but their performance may suffer from excessive pessimism. Advantage-weighted regression [35, 33, 45] improves upon behavior policy, while simultaneously enforcing an implicit KL-divergence constraint. Recent sequence modeling methods based on Transformers [43] also show competitive performance in both model-free [5] or model-based [14] paradigm.

## 3   Background

The reinforcement learning problem [41] is formulated as decision making in the environment represented by a Markov Decision Process (MDP) $\mathcal{M} = (\mathcal{S}, \mathcal{A}, \rho_0, p, r, \gamma)$, where $\mathcal{S}$ is the state space, $\mathcal{A}$ is the action space, $\rho_0(s_0)$ is the initial state distribution, $p(s'|s, a)$ is the transition distribution, $r(s, a)$ is the reward function, and $\gamma$ is the discount factor. The goal in RL is to find a policy $\pi(a|s)$ maximizing the expected return: $\mathbb{E}_\pi \left[ \sum_{t=0}^\infty \gamma^t r(s_t, a_t) \right]$.

The optimal state-action value function or Q function $Q^*(s, a)$ measures the expected return starting in state $s$ taking action $a$ and then acting optimally thereafter. A corresponding optimal policy can be obtained through greedy action choices $\pi^*(s) = \arg\max_a Q^*(s, a)$. The Q-learning algorithm [46] learns $Q^*$ via iterating the Bellman optimality operator $\mathcal{T}$, defined as: $\mathcal{T}Q(s, a) = \mathbb{E}_{s'} \left[ r(s, a) + \gamma \max_{a'} Q(s', a') \right]$. For large or continuous state space, the value can be represented by function approximators $Q_\theta(s, a)$ with parameters $\theta$. In practice, the parameters $\theta$ are updated by minimizing the mean squared Bellman error with an experience replay dataset $\mathcal{D}$ and a target function $Q_{\bar{\theta}}$ [32]: $J_Q(\theta) = \mathbb{E}_{(s,a,r,s') \sim \mathcal{D}} \left[ Q_\theta(s, a) - r - \gamma \max_{a'} Q_{\bar{\theta}}(s', a') \right]^2$.

In a continuous action space, the analytic maximum is intractable. Actor-Critic methods [41, 8, 12] perform action selection with a separate policy function $\pi_\phi$ maximizing the value function:

$$J_Q(\theta) = \mathbb{E}_{(s,a,r,s') \sim \mathcal{D}} \left[ Q_\theta(s, a) - r - \gamma Q_{\bar{\theta}}(s', \pi_\phi(s')) \right]^2. \tag{1}$$

The policy can be updated following the deterministic policy gradient (DPG) theorem [39]:

$$J_\pi(\phi) = \mathbb{E}_{s \sim \mathcal{D}} \left[ -Q_\theta(s, \pi_\phi(s)) \right]. \tag{2}$$

### 3.1   Offline Reinforcement Learning

In contrast to online RL methods, which interact with environment to collect experience data, offline RL [28, 29] methods learn from a finite and fixed dataset $\mathcal{D} = \{(s, a, r, s')\}$ which has been collected by some unknown behavior policy $\pi_\beta$. Direct application of off-policy methods on offline setting suffers from *extrapolation error* [9, 24], which means that an *out-of-distribution* action $a$ in state $s$ can produce erroneously overestimated values $Q_\theta(s, a)$.

## 3.2 Support Constraint in Offline RL

To avoid *out-of-distribution* actions from function approximation, support constraint [9, 24, 11] is commonly used, which means in state $s$ to only allow action that has $\epsilon$-support under behavior policy: $\{a : \pi_\beta(a|s) > \epsilon\}$. Kumar *et al.* [24] first introduce the *distribution-constrained operator*, which is instantiated to the *supported operator* in this work:

**Definition 3.1.** Given behavior policy $\pi_\beta$ and threshold $\epsilon$, the *supported backup operator* is defined as

$$\mathcal{T}_\epsilon Q(s,a) = \mathbb{E}_{s'} \left[ r(s,a) + \gamma \max_{a':\pi_\beta(a'|s')>\epsilon} Q(s',a') \right] \tag{3}$$

with its fixed point $Q_\epsilon^*(s,a)$ named as the *supported optimal Q function*.

Following the theoretical analysis of Kumar *et al.* [24], we can obtain a bound of how suboptimal $Q_\epsilon^*$ may be with respect to the optimal Q function $Q^*$:

**Corollary 3.2.** *Let* $\alpha(\epsilon) = \|\mathcal{T}Q^* - \mathcal{T}_\epsilon Q^*\|_\infty = \max_{s,a} |\mathcal{T}Q^*(s,a) - \mathcal{T}_\epsilon Q^*(s,a)|$. *The suboptimality of* $Q_\epsilon^*(s,a)$ *can be upper-bounded as* $\|Q^* - Q_\epsilon^*\|_\infty \leq \frac{1}{1-\gamma}\alpha(\epsilon)$.

Note that the complete theoretical results from Kumar *et al.* [24] also includes a term w.r.t. the bootstrapping error. We refer readers to Kumar *et al.* [24] for more details. Similar results are also presented by Ghasemipour *et al.* [11].

## 4 Supported Policy Optimization

As aforementioned, performing support constraint is the typical method to mitigate extrapolation error in offline RL. Noticing that the support constraint can be formalized based on the density of behavior policy, we propose the **S**upported **P**olicy **O**p**T**imization (SPOT) as a regularization method from the new perspective of explicit density estimation. Concretely, SPOT involves a new regularization term, which is directly derived from the theoretical formalization of the support constraint. Besides, a conditional VAE is adopted to explicitly estimate the behavior density in the regularization term. Plugged into off-policy RL algorithms, we will finally arrive at the practical algorithm of SPOT.

### 4.1 Support Constraint via Behavior Density

Similar to how the optimal policy can be extracted from the optimal Q function, the *supported optimal policy* can also be recovered via greedy selection: $\pi_\epsilon^*(s) = \arg\max_{a:\pi_\beta(a|s)>\epsilon} Q_\epsilon^*(s,a)$. For the case of function approximation, it corresponds to a constrained policy optimization problem.

While prior works use specific parameterization of $\pi$ [9, 11] or divergence penalty [24, 47] to perform support constraint, we propose to directly use behavior density $\pi_\beta(\cdot|s)$ as constraint[1]:

$$\max_\phi \; \mathbb{E}_{s\sim\mathcal{D}} [Q_\theta(s,\pi_\phi(s))]$$
$$\text{s.t.} \; \min_s \; \log \pi_\beta(\pi_\phi(s)|s) > \hat{\epsilon}, \tag{4}$$

where $\hat{\epsilon} = \log\epsilon$ for notational simplicity. Constraint via behavior density is simple and straightforward in the context of support constraint. We adopt log-likelihood instead of raw likelihood because of its mathematical convenience.

This problem imposes a constraint that the density of behavior policy is lower-bounded at every point in the state space, which is impractical to solve due to the large even infinite number of constraints. Following previous works from both online RL [38] and offline RL [24, 35] w.r.t. constrained policy optimization, we instead use a heuristic approximation that considers the average behavior density:

$$\max_\phi \; \mathbb{E}_{s\sim\mathcal{D}} [Q_\theta(s,\pi_\phi(s))]$$
$$\text{s.t.} \; \mathbb{E}_{s\sim\mathcal{D}} [\log \pi_\beta(\pi_\phi(s)|s)] > \hat{\epsilon}. \tag{5}$$

---

[1]With slight abuse of notation, here $\pi_\beta(\cdot|s)$ stands for an action distribution of the stochastic policy, while $\pi_\phi(s)$ stands for a deterministic action taken by the policy.

Converting the constrained optimization problem into an unconstrained one by treating the constraint term as a penalty, we finally get the policy learning objective as

$$J_\pi(\phi) = \mathbb{E}_{s \sim \mathcal{D}} \left[ -Q_\theta(s, \pi_\phi(s)) - \lambda \log \pi_\beta(\pi_\phi(s)|s) \right], \tag{6}$$

where $\lambda$ is a Lagrangian multiplier.

## 4.2 Explicit Estimation of Behavior Density

The straightforward regularization term in Eq. (6) requires access to $\pi_\beta$. While we only have offline data generated by $\pi_\beta$, we can explicitly estimate the probability density at an arbitrary point with the density estimation methods [1].

The variational autoencoder (VAE) [19] is among the best performing neural density-estimation models and we opt to use a conditional variational autoencoder [40] as our density estimator. Typically, $\pi_\beta(a|s)$ can be approximated by a Deep Latent Variable Model $p_\psi(a|s) = \int p_\psi(a|z,s)p(z|s)\mathrm{d}z$ with a fixed prior $p(z|s) = \mathcal{N}(\mathbf{0}, I)$. While the marginal likelihood $p_\psi(a|s)$ is intractable, VAE additionally uses an approximate posterior $q_\varphi(z|a,s) \approx p_\psi(z|a,s)$ and parameters $\psi$ and $\varphi$ can be optimized jointly with evidence lower bound (ELBO):

$$
\begin{aligned}
\log p_\psi(a|s) &\geq \mathbb{E}_{q_\varphi(z|a,s)} \left[ \log \frac{p_\psi(a,z|s)}{q_\varphi(z|a,s)} \right] \\
&= \mathbb{E}_{q_\varphi(z|a,s)} \left[ \log p_\psi(a|z,s) \right] - \mathrm{KL}\left[ q_\varphi(z|a,s) \| p(z|s) \right] \\
&\stackrel{\mathrm{def}}{=} -\mathcal{L}_{\mathrm{ELBO}}(s, a; \varphi, \psi).
\end{aligned}
\tag{7}
$$

After training a VAE, we can simply use $-\mathcal{L}_{\mathrm{ELBO}}$ to lower-bound $\log p_\psi(a|s)$ and thus approximately lower-bound $\log \pi_\beta$ in Eq. (6). However, there theoretically exists a bias between them, as we know $\log p_\psi(a|s) = -\mathcal{L}_{\mathrm{ELBO}} + \mathrm{KL}(q_\varphi(z|a,s)\|p_\psi(z|a,s))$. To obtain an estimation with lower bias, we can use the importance sampling technique [37, 20]:

$$
\begin{aligned}
\log p_\psi(a|s) &= \log \mathbb{E}_{q_\varphi(z|a,s)} \left[ \frac{p_\psi(a,z|s)}{q_\varphi(z|a,s)} \right] \\
&\approx \mathbb{E}_{z^{(l)} \sim q_\varphi(z|a,s)} \left[ \log \frac{1}{L} \sum_{l=1}^{L} \frac{p_\psi(a, z^{(l)}|s)}{q_\varphi(z^{(l)}|a,s)} \right] \\
&\stackrel{\mathrm{def}}{=} \widehat{\log \pi_\beta}(a|s; \varphi, \psi, L).
\end{aligned}
\tag{8}
$$

Burda *et al.* [3] show that $\widehat{\log \pi_\beta}(a|s; \varphi, \psi, L)$ gives a lower bound of $\log p_\psi(a|s)$ and the bound becomes tighter as $L$ increases. Note that here we adopt sampling of VAE to directly estimate the density of the behavior policy instead of to estimate the divergence [24].

In summary, the loss function in Eq. (6) can be implemented with the explicit density estimator as follows:

$$J_\pi(\phi) = \mathbb{E}_{s \sim \mathcal{D}} \left[ -Q_\theta(s, \pi_\phi(s)) - \lambda \widehat{\log \pi_\beta}(\pi_\phi(s)|s; \varphi, \psi, L) \right]. \tag{9}$$

## 4.3 Practical Algorithm

The general framework derived above can be built on top of off-policy algorithms with minimal modifications. We choose TD3 [8] as our base algorithm, which recently shows strong resistance to overestimation in offline RL [7] (see Section 5.2 for a detailed discussion).

**Q normalization.** Following TD3+BC [7], we add a normalization term to policy loss as a default option for better balance between Q value objective and regularization: $J_\pi(\phi) = \mathbb{E}_{s \sim \mathcal{D}} \left[ \frac{-Q_\theta(s, \pi_\phi(s))}{\alpha} - \lambda \widehat{\log \pi_\beta}(\pi_\phi(s)|s; \varphi, \psi, L) \right]$, where $\alpha = \frac{1}{N} \sum_{s_i} |Q(s_i, \pi_\phi(s_i))|$ is the normalization term based on the minibatch $\{s_i\}_{i=1}^N$ with size $N$.

**Simpler density estimator.** While $\widehat{\log \pi_\beta}(a|s; \varphi, \psi, L)$ with large $L$ is much tighter, we empirically find there is no further improvement with larger $L$ compared to $L = 1$ (see Figure 7 in Appendix for results of ablation study). To make our algorithm simpler, we choose to only use $L = 1$ for

---

**Algorithm 1** Supported Policy Optimization (SPOT)

---

**Input:** Dataset $\mathcal{D} = \{(s, a, r, s')\}$
// **VAE Training**
Initialize VAE with parameters $\psi$ and $\varphi$
**for** $t = 1$ **to** $T_1$ **do**
    Sample minibatch of transitions $(s, a) \sim \mathcal{D}$
    Update $\psi, \varphi$ minimizing $\mathcal{L}_{\mathrm{ELBO}}(s, a; \varphi, \psi)$ in Eq. (7)
**end for**
// **Policy Training**
Initialize the policy network $\pi_\phi$, critic network $Q_\theta$ and target network $Q_{\bar{\theta}}$ with $\bar{\theta} \leftarrow \theta$
**for** $t = 1$ **to** $T_2$ **do**
    Sample minibatch of transitions $(s, a, r, s') \sim \mathcal{D}$
    Update $\theta$ minimizing $J_Q(\theta)$ in Eq. (1)
    Update $\phi$ minimizing $J_\pi(\phi)$ in Eq. (9)
    Update target network: $\bar{\theta} \leftarrow \tau\theta + (1 - \tau)\bar{\theta}$
**end for**

---

a practical estimator, which is just the ELBO estimator of the VAE: $\widehat{\log \pi_\beta}(a|s; \varphi, \psi, L = 1) = -\mathcal{L}_{\mathrm{ELBO}}(s, a; \varphi, \psi)$. Note that with $L = 1$, we can analytically separate out the KL divergence as Eq. (7) to enjoy a simpler and lower-variance update.

**Overall algorithm.** Putting everything together, the full algorithm is summarized in Algorithm 1. Our algorithm first trains VAE using $\mathcal{L}_{\mathrm{ELBO}}(s, a; \varphi, \psi)$ to obtain a density estimator with sufficient accuracy. Then it turns to policy training analogous to common Actor-Critic methods except that we plug the regularization term computed by the density estimator into the policy loss $J_\pi(\phi)$.

## 5 Experimental Evaluation

Our experiments aim to evaluate our method comparatively, in contrast to prior offline RL methods, focusing on both offline training and online fine-tuning. We first demonstrate the effect of $\lambda$ on applying support constraint and show that our method is able to learn a policy with the strongest performance at the same level of constraint strength, compared to previous policy constraint methods. We then evaluate SPOT on D4RL benchmark [6], studying how effective our method is in contrast to a broader range of state-of-the-art offline RL methods. Finally, we study how SPOT compares to prior methods when fine-tuning with online RL from an offline RL initialization, and investigate the computational efficiency of different methods. Code is available at `https://github.com/thuml/SPOT`.

### 5.1 Analysis of Support Constraint in SPOT

**Effect of $\lambda$ on constraint strength.** The coefficient $\lambda$ in SPOT is essential and corresponds to a specific constraint strength in the constrained policy optimization problem formalized in Eq. (4). To illustrate how $\lambda$ effects the learned policy, we evaluate behavior density of actions taken by the policy learned with varying values of $\lambda \in [0.05, 0.1, 0.2, 0.5]$ on standard D4RL [6] Gym-MuJoCo domains. Concretely, we plot the distribution of behavior density $\log \pi_\beta(\pi_\phi(s)|s), s \sim \mathcal{D}$ in Figure 1a, where $\log \pi_\beta$ is estimated by our learned density estimator (Eq. (8)) with $L$ set to a sufficient large number 500 for more accurate estimation. As we show, with smaller $\lambda$, the learned policy is much more possible to perform actions with low behavior density $\log \pi_\beta(\pi_\phi(s)|s)$. On the other hand, policies learned by higher $\lambda$ are restricted to take only high-density actions.

**Tradeoff between constraint strength and optimality.** It has been shown by Kumar *et al.* [24] that the optimality of approximate supported optimal policy is lower-bounded by a tradeoff between keeping the learned policy supported by the behavior policy (controlling extrapolation error) and keeping the supported policy set large enough to capture well-performing policies. If the constraint in Eq. (4) is strong (by a large $\log \epsilon$), the extrapolation error is restrained to be small but the optimal

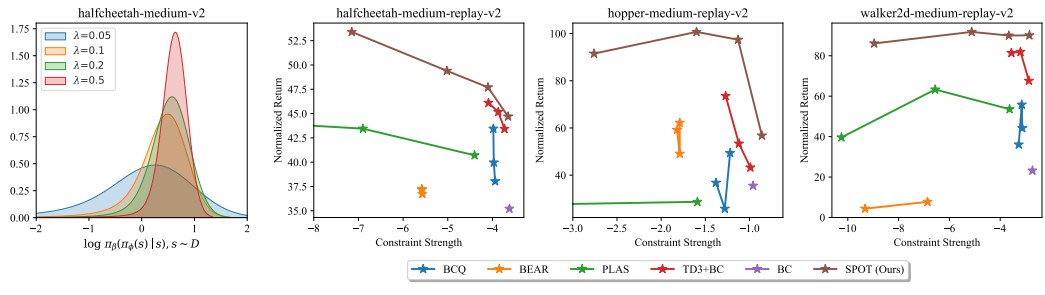

(a) Effect of $\lambda$.      (b) Tradeoff between constraint strength and optimality.

Figure 1: Analysis on constraint strength: (a) With varying values of the coefficient $\lambda$, SPOT applies support constraint with different strength, which is demonstrated by the behavior density of actions taken by the learned policy: $\log \pi_\beta(\pi_\phi(s)|s), s \sim \mathcal{D}$. (b) When evaluating the performance at varying levels of constraint strength, SPOT takes shape of the "upper envelope" of all methods, showing that SPOT can always achieve the strongest performance among extensive policy constraint methods. Constraint strength is captured approximately by the 5th-percentile of the distribution $\log \pi_\beta(\pi_\phi(s)|s), s \sim \mathcal{D}$. Extended results can be found in Appendix C.4.

policy under constraint may have poor performance. Otherwise, if the constraint is weak, well-performing policies can be learned though at the risk of the extrapolation error.

We aim to answer the question that *at the same level of constraint being satisfied, is SPOT able to learn a policy with the strongest performance compared to previous policy constraint methods?* We compare SPOT with BC [36], BCQ [9], BEAR [24], PLAS [50] and TD3+BC [7]. Hyperparameters to control constraint strength of various methods are adjusted to several values to form a spectrum of constraint strength (see Appendix C.4 for details). We approximate the satisfied constraint strength $\log \epsilon$ by the 5th-percentile of the distribution $\log \pi_\beta(\pi_\phi(s)|s), s \sim \mathcal{D}$. As shown in Figure 1b, our method SPOT takes shape of the "upper envelope" of all methods, demonstrating that taking advantage of exact standard formulation of support constraint, SPOT is flexible to learn the supported optimal policy and resist extrapolation error at the same time. Note that the divergence-based regularization method BEAR [24] yields a poor performance in our experiments and only satisfies a loose constraint in contrast to other baselines, showing that it cannot prevent out-of-distribution actions effectively and suffers from extrapolation error with indirect divergence regularization.

### 5.2 Comparisons on Offline RL Benchmarks

Next, we evaluate our approach on the D4RL benchmark [6] in comparison to state-of-the-art methods. We focus on Gym-MuJoCo locomotion domains and much more challenging AntMaze domains, which consists of sparse-reward tasks and requires "stitching" fragments of suboptimal trajectories traveling undirectedly in order to find a path from the start to the goal of the maze.

**Baselines.** We select the classic BC [36] and state-of-the-art offline RL methods as baselines. For methods based on dynamic programming, we compare to AWAC [33], Onestep RL [2], TD3+BC [7], CQL [26], and IQL [23]. We also include the sequence-modeling method Decision Transformer [5].

**Hyperparameter tuning.** The weight $\lambda$ in Eq. (9) is essential for policy constraint. Following prior works [2, 24, 47], we allow access to the online environment to tune a small set of the hyperparameter $\lambda$ ($< 10$ choices) , which is a reasonable setup for practical applications. See Appendix C.1 for additional discussion and details.

**Gym-MuJoCo domains.** Results for the Gym-MuJoCo domains are shown in Table 2. As we show, SPOT substantially outperforms state-of-the-art methods, especially in suboptimal "medium" and "medium-replay" datasets with a large margin, which further demonstrates the advantages of direct constraint on behavior density proposed by SPOT.

**AntMaze domains.** Results for the AntMaze domains are shown in Table 3. Note that D4RL recently releases a bug-fixed "-v2" version of AntMaze datasets, and thus we select competitive baselines and rerun their author-provided implementations for comparison. We also include results for BCQ and

Table 2: Performance of SPOT and prior methods on Gym-MuJoCo tasks. m = medium, m-r = medium-replay, m-e = medium-expert. For baselines, we report numbers directly from the IQL paper [23], which provides a unified comparison for "-v2" datasets. For SPOT, we report the mean and standard deviation for 10 seeds.

|  | BC | AWAC | DT | Onestep | TD3+BC | CQL | IQL | SPOT (Ours) |
|---|---|---|---|---|---|---|---|---|
| HalfCheetah-m-e-v2 | 55.2 | 42.8 | 86.8 | **93.4** | 90.7 | 91.6 | 86.7 | 86.9±4.3 |
| Hopper-m-e-v2 | 52.5 | 55.8 | **107.6** | 103.3 | 98.0 | 105.4 | 91.5 | 99.3±7.1 |
| Walker-m-e-v2 | 107.5 | 74.5 | 108.1 | **113.0** | 110.1 | 108.8 | 109.6 | 112.0±0.5 |
| HalfCheetah-m-v2 | 42.6 | 43.5 | 42.6 | 48.4 | 48.3 | 44.0 | 47.4 | **58.4**±1.0 |
| Hopper-m-v2 | 52.9 | 57.0 | 67.6 | 59.6 | 59.3 | 58.5 | 66.2 | **86.0**±8.7 |
| Walker-m-v2 | 75.3 | 72.4 | 74.0 | 81.8 | 83.7 | 72.5 | 78.3 | **86.4**±2.7 |
| HalfCheetah-m-r-v2 | 36.6 | 40.5 | 36.6 | 38.1 | 44.6 | 45.5 | 44.2 | **52.2**±1.2 |
| Hopper-m-r-v2 | 18.1 | 37.2 | 82.7 | 97.5 | 60.9 | 95.0 | 94.7 | **100.2**±1.9 |
| Walker-m-r-v2 | 26.0 | 27.0 | 66.6 | 49.5 | 81.8 | 77.2 | 73.8 | **91.6**±2.8 |
| Gym-MuJoCo total | 466.7 | 450.7 | 672.6 | 684.6 | 677.4 | 698.5 | 692.4 | **773.0**±30.2 |

Table 3: Performance of SPOT and prior methods on AntMaze tasks. For baselines, we obtain the results using author-provided implementations on "-v2" datasets. For BCQ and BEAR, we report numbers from D4RL paper [6]. For SPOT, we report the mean and standard deviation for 10 seeds.

|  | BCQ | BEAR | BC | DT | TD3+BC | PLAS | CQL | IQL | SPOT (Ours) |
|---|---|---|---|---|---|---|---|---|---|
| umaze-v2 | 78.9 | 73.0 | 49.2 | 54.2±4.1 | 73.0±34.0 | 62.0±16.7 | 82.6±5.7 | 89.6±4.2 | **93.5**±2.4 |
| umaze-diverse-v2 | 55.0 | 61.0 | 41.8 | 41.2±11.4 | 47.0±7.3 | 45.4±7.9 | 10.2±6.7 | **65.6**±8.3 | 40.7±5.1 |
| medium-play-v2 | 0.0 | 0.0 | 0.4 | 0.0±0.0 | 0.0±0.0 | 31.4±21.5 | 59.0±1.6 | **76.4**±2.7 | 74.7±4.6 |
| medium-diverse-v2 | 0.0 | 8.0 | 0.2 | 0.0±0.0 | 0.2±0.4 | 20.6±27.7 | 46.6±24.0 | 72.8±7.0 | **79.1**±5.6 |
| large-play-v2 | 6.7 | 0.0 | 0.0 | 0.0±0.0 | 0.0±0.0 | 2.2±3.8 | 16.4±17.1 | **42.0**±3.8 | 35.3±8.3 |
| large-diverse-v2 | 2.2 | 0.0 | 0.0 | 0.0±0.0 | 0.0±0.0 | 3.0±6.7 | 3.2±4.1 | **46.0**±4.5 | 36.3±13.7 |
| AntMaze total | 142.8 | 142.0 | 91.6 | 95.4±15.5 | 120.2±41.7 | 164.6±84.3 | 218.0±59.2 | **392.4**±30.5 | 359.6±39.7 |

BEAR trained on "-v0" datasets, directly from [6]. It is not suitable to compare them with reproduced baselines but we include them in order to highlight that previous policy constraint methods struggle to succeed in training on challenging AntMaze domains.

As shown in Table 3, SPOT performs slightly worse than IQL but outperforms remaining modern offline RL baselines, including the pessimistic value method CQL and the sequence modeling method Decision Transformer. Note that IQL ingeniously designs multi-step dynamic programming and policy extraction steps to apply an implicit constraint for offline RL, but when online fine-tuned after offline RL initialization, IQL is inferior to SPOT. SPOT's pluggable design can take full advantage of existing online RL algorithms (see Section 5.3 and Table 4). To the best of our knowledge, our algorithm is the first to train successfully in challenging AntMaze domains with pluggable modification on top of off-policy RL methods for offline RL.

**Ablation.** In Figure 2, we perform an ablation study over the components in our method. First, we replace the VAE-based density estimator with a behavioral-cloned Gaussian policy model [24, 47]. As expected, it degrades the performance due to the lack of flexibility to model complex distribution, especially on Gym-MuJoCo medium-expert datasets and AntMaze datasets. Then, we evaluate SPOT without Q normalization and find that the removal provides some benefits as well as damages on different datasets, with only an insignificant impact on total performance. Nevertheless, we include it as a default option following TD3+BC [7].

Lastly, we investigate how the base off-policy algorithm matters. Since SAC [12] has similar off-policy performance with TD3, we attempt to adopt SAC as our base method. As Xu *et al.* [48] argue that the entropy term will do harm to the offline setting, we also implement a variant of SAC without maximum entropy regularization. Surprisingly, both variants are vulnerable to pathological extrapolation error on AntMaze domains and provide poor performance (see Figure 2). We argue that

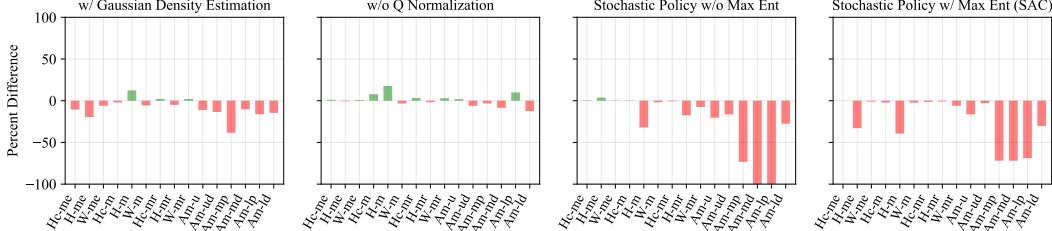

Figure 2: Percent difference of the performance of an ablation of SPOT, compared with the original algorithm. Hc = HalfCheetah, H = Hopper, W = Walker, me = medium-expert, m = medium, mr = medium-replay. Am = AntMaze, u = umaze, m = medium, l = large, d = diverse, p = play. Implementation details and quantitative results can be found in Appendix C.2.

several key native designs of TD3 are beneficial to offline RL. **(1).** For the case of learning a stochastic policy *without* entropy regularization, we find that the learned policy quickly degenerates into a deterministic one (with near-zero standard deviation). A concern with deterministic policies is that they are prone to overfit overestimated actions and propagate the estimation error through Bellman backups, which is even serious in offline RL. Hopefully, TD3 introduces Target Policy Smoothing [8, 42] into Eq. (1), which adds random noise to target actions and can alleviate the effect of error propagation. **(2).** For the case *with* entropy regularization, Bellman backup of a stochastic policy resembles Target Policy Smoothing, then the primary distinction may come from the actor learning objective. While TD3 produces a deterministic action minimizing Eq. (9), stochastic policies are more likely to produce out-of-distribution actions with erroneous estimated values, so the policy gradient may become biased as well as with high variance. On much easier Gym-MuJoCo domains, SAC-style SPOT variants are comparable to the TD3 variant on most of the datasets, but we remark that TD3 indeed extends the limit of policy constraint methods on some tasks, such as hopper-medium. Our analysis suggests that TD3 may be preferable for offline RL as the base off-policy method with native designs (such as "stochastic" critic training and deterministic actor training) addressing function approximation error not only in the online setting but also in the offline setting [7].

## 5.3 Online Fine-tuning after Offline RL

Pluggable SPOT can be online fine-tuned seamlessly, which means that we only need to gradually decay the constraint strength $\lambda$ in the online phase in order to avoid excessive conservatism. As AWAC [33] shows that behavior models are hard to update online, we fix the VAE during online fine-tuning. Note that when $\lambda$ is zero, our algorithm is exactly the standard off-policy RL algorithm that SPOT builds upon. It is beneficial since we enjoy a minimal gap between offline RL and well-established online RL methods and can take full advantage of them for online fine-tuning.

Since IQL [23] is the strongest baseline in our offline experiments, which also has shown superior online performance than prior methods [33, 26] in its paper, and most of the other baselines fail to learn meaningful results, we follow the experiments of IQL and compare to IQL in online fine-tuning. We also compare to our base RL method TD3 [8] trained online from scratch. We use the challenging AntMaze domains [6]. During online fine-tuning of SPOT, the regularization weight $\lambda$ is linearly decayed to one-fifth of its initial value. See Appendix C.3 for details.

Table 4: Online fine-tuning results on AntMaze tasks, showing initial performance after offline RL and performance after 1M steps of online RL. All numbers are reported by the mean of 5 seeds.

| | IQL | SPOT (Ours) |
|---|---|---|
| umaze-v2 | $85.4 \rightarrow 96.2$ | **93.2** $\rightarrow$ **99.2** (+3.0) |
| umaze-diverse-v2 | **70.8** $\rightarrow$ 62.2 | $41.6 \rightarrow$ **96.0** (+33.8) |
| medium-play-v2 | $68.6 \rightarrow 89.8$ | **75.2** $\rightarrow$ **97.4** (+7.6) |
| medium-diverse-v2 | **73.4** $\rightarrow$ **90.2** | $73.0 \rightarrow$ **96.2** (+6.0) |
| large-play-v2 | $40.0 \rightarrow 78.6$ | **40.8** $\rightarrow$ **89.4** (+10.8) |
| large-diverse-v2 | $40.4 \rightarrow 73.4$ | **44.0** $\rightarrow$ **90.8** (+17.4) |
| AntMaze total | **378.6** $\rightarrow$ 490.4 | $367.8 \rightarrow$ **569.0** (+78.6) |

Results are shown in Table 4. While online training from scratch fails in the challenging sparse reward tasks on AntMaze domains, SPOT initialized with offline RL succeeds to learn nearly optimal

policies and performs significantly better than the strongest baseline IQL, especially in the most difficult large maze.

## 5.4 Computation Cost

Regularization methods, including our SPOT, benefiting from the pluggable design, only need one forward pass of the policy network to do inference, while parameterization methods always need inference through secondary components, such as the generative model or the critic network, which may bring extra time or memory cost. As demonstrated empirically in Figure 3, parameterization methods are usually slower than regularization methods. TD3+BC and our SPOT run more than two times faster compared to the most time-consuming BCQ, while SPOT also has the advantage of parameterization methods, which explicitly constrain the behavior density of learned actions.

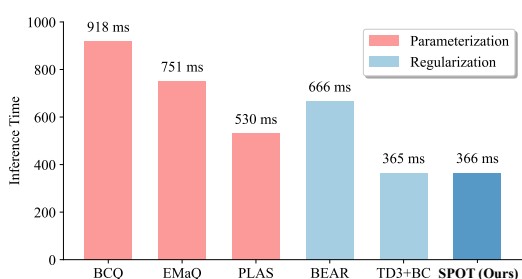

Figure 3: Runtime of various offline RL algorithms interacting with the HalfCheetah environment to produce a 1000-steps trajectory. See Appendix C.5 for the details.

SPOT indeed adds some training overhead due to the VAE-based density estimator, but is still much simpler than most methods. A comparison of training time is provided in Appendix C.5. SPOT lies in the second tier, only slightly worse than PLAS and beaten by the simplest TD3+BC.

## 6 Conclusion

We present Supported Policy OpTimization (SPOT), a policy constraint method to offline RL built upon off-the-shelf off-policy RL algorithms. SPOT introduces a pluggable regularization term applied directly to the estimated behavior density, which brings a number of important benefits. First, our algorithm is computationally efficient at inference, which only needs one forward process of the policy network for action selection. Second, capturing the standard formulation of the support constraint based on behavior density, it obtains excellent performance across different tasks in the D4RL benchmarks, including standard Gym-MuJoCo tasks and much more challenging AntMaze tasks. Finally, the pluggable design of our algorithm makes it seamless to apply online fine-tuning after offline RL pre-training. Taking full advantage of well-established online methods, SPOT exceeds the state-of-the-art online fine-tuning performance on the challenging AntMaze domains.

**Limitations.** One limitation of our current method, shared by most policy constraint methods, is that the performance may be limited by the accuracy of estimation of the behavior policy. Advances in generative models, such as diffusion models [13, 44, 4], may improve the real-world performance of offline RL, especially in scenarios with highly multimodal behaviors. An exciting direction for future work would be to develop an effective pluggable constraint mechanism excluding explicit estimation of behavior policy. Another limitation of our work is that we rely on online evaluation to select the best set of hyperparameters. Although this evaluation protocol is commonly adopted by the literature of offline RL, extensive online evaluation is not practical in real-world applications and online evaluation budgets may have a significant impact on final performance [21, 27]. Specifically for SPOT, the selection of regularization weight without requirements for extensive online evaluation is critical and needs to be developed, either by offline policy evaluation [34], by manual tuning based on offline training metrics and conditions [25], or by auto-tuning with a dual optimization [12].

## Acknowledgments

We would like to thank many colleagues, in particular, Jincheng Zhong, Haoyu Ma, Yiwen Qiu, and Yuchen Zhang for their valuable discussion and support for this work. This work was supported by the National Key Research and Development Plan (2020AAA0109201), National Natural Science Foundation of China (62022050 and 62021002), Beijing Nova Program (Z201100006820041), and BNRist Innovation Fund (BNR2021RC01002).

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
