# Supplementary Material: *Supported Policy Optimization for Offline Reinforcement Learning*

**Jialong Wu**[1], **Haixu Wu**[1], **Zihan Qiu**[2], **Jianmin Wang**[1], **Mingsheng Long**[1]✉

[1]School of Software, BNRist, Tsinghua University, China
[2]Institute for Interdisciplinary Information Sciences, Tsinghua University, China
{wujialong0229,qzh11628}@gmail.com, whx20@mails.tsinghua.edu.cn
{jimwang,mingsheng}@tsinghua.edu.cn

## A  Proofs

### A.1  Proof of Corollary 3.2

This proof is adapted from the proof of Theorem 4.1 in Kumar *et al.* [11].

*Proof.*

$$
\begin{aligned}
\|Q^* - Q_\epsilon^*\|_\infty &= \|\mathcal{T}Q^* - \mathcal{T}_\epsilon Q_\epsilon^*\|_\infty \\
&\leq \|\mathcal{T}_\epsilon Q^* - \mathcal{T}_\epsilon Q_\epsilon^*\|_\infty + \|\mathcal{T}Q^* - \mathcal{T}_\epsilon Q^*\|_\infty \\
&\leq \gamma \|Q^* - Q_\epsilon^*\|_\infty + \alpha(\epsilon)
\end{aligned}
\tag{1}
$$

Thus we have $\|Q^* - Q_\epsilon^*\|_\infty \leq \frac{1}{1-\gamma}\alpha(\epsilon)$. □

## B  Missing Background: Parameterization Methods for Policy Constraint

Parameterization methods enforce the learned policy $\pi$ to be close to the behavior policy $\pi_\beta$ with various specific parameterization of $\pi$.

**BCQ** [6] learns a generative model of behavior policy $\pi_\beta$ and trains the actor as a perturbation model $\xi_\phi$ to perturb the randomly generated actions. The policy parameterized by BCQ is to greedily select the one maximizing Q function among a large number $N$ of perturbed sampled actions $a_i + \xi_\phi(s, a_i)$:

$$
\pi(a|s) := \underset{a_i + \xi_\phi(s, a_i)}{\arg\max}\, Q_\theta(s, a_i + \xi_\phi(s, a_i)) \text{ for } a_i \sim \pi_\beta(a|s), i = 1, \ldots, N.
\tag{2}
$$

**EMaQ** [7] simplifies BCQ by discarding the perturbation models:

$$
\pi(a|s) := \underset{a_i}{\arg\max}\, Q_\theta(s, a_i) \text{ for } a_i \sim \pi_\beta(a|s), i = 1, \ldots, N.
\tag{3}
$$

**PLAS** [17] learns a policy $z = \pi_\phi(s)$ in the latent space of the generative model and parameterizes the policy using the decoder $D_\beta : z \mapsto a$ of the generative model on top of the latent policy:

$$
\pi(a|s) := D_\beta(\pi_\phi(s)).
\tag{4}
$$

## C  Implementation Details and Extended Results

### C.1  Benchmark Experiments (Table 2 and 3)

**Data.** We use the datasets from the D4RL benchmark [3], of the latest versions, which are "v2" for both Gym-MuJoCo and AntMaze domains.

36th Conference on Neural Information Processing Systems (NeurIPS 2022).

**Baselines.** We report Gym-MuJoCo results of baselines directly from IQL paper [10] and rerun competitive baselines for AntMaze tasks on "v2" datasets, taking their official implementations:

- BC [15]: modified from `https://github.com/sfujim/TD3_BC`
- Decision Transformer [2]: `https://github.com/kzl/decision-transformer`
- TD3+BC [5]: `https://github.com/sfujim/TD3_BC`
- CQL [12]: `https://github.com/aviralkumar2907/CQL`
- IQL [10]: `https://github.com/ikostrikov/implicit_q_learning/`

**Implementation details.** Our algorithm SPOT consists of two stages, namely VAE training and policy training. We will introduce the details of both stages respectively.

For VAE training, our code is based on the implementation of BCQ: `https://github.com/sfujim/BCQ/tree/master/continuous_BCQ`. Following TD3+BC [5], we normalize the states in the dataset for Gym-MuJoCo domains but do not normalize the states for AntMaze domains. Hyperparameters used by VAE are in Table 5.

Table 5: Hyperparameters of VAE training in SPOT.

|  | Hyperparameter | Value |
|---|---|---|
| VAE training | Optimizer | Adam [9] |
|  | Learning rate | $1 \times 10^{-3}$ |
|  | Batch size | 256 |
|  | Number of iterations | $10^5$ |
|  | KL term weight | 0.5 |
|  | Normalized states | *True* for Gym-MuJoCo |
|  |  | *False* for AntMaze |
| VAE architecture | Encoder hidden dim | 750 |
|  | Encoder layers | 3 |
|  | Latent dim | $2 \times$ action dim |
|  | Decoder hidden dim | 750 |
|  | Decoder layers | 3 |

For policy training, our code is based on the implementation of TD3+BC [5]. Following IQL [10], we subtract 1 from rewards for the AntMaze datasets. Hyperparameters used by policy training are in Table 6.

For evaluation, we average mean returns overs 10 evaluation trajectories on the Gym-MuJoCo tasks, and average over 100 evaluation trajectories on the AntMaze tasks.

**Hyperparameter tuning.** The weight of regularization term $\lambda$ is essential for SPOT to control different strengths of the constraint. Following prior works [1, 11, 16], we allow access to the environment to tune a small ($< 10$) set of the hyperparameter $\lambda$. The hyperparameter sets for Gym-MuJoCo and AntMaze domains can be seen in Table 6. We tune hyperparameters using 3 seeds but then evaluate the best hyperparameter by training with totally new 10 seeds and then report final results on these additional 10 seeds.

As discussed in Brandfonbrener *et al.* [1], it's a reasonable setup for applications like robotics, where we can test a limited number of trained policies on a real system. Beyond the scope of this work, we believe that to make offline RL more widely applicable, better approaches for offline policy evaluation and selection are indispensable. Fortunately, it has attracted wide attention by the community and we refer the reader to [14, 4]. Note that Paine *et al.* [14] demonstrate that policies learned by policy constraint methods have the advantage of being easier to evaluate and rank offline, since the methods encourage learned policies to stay close to the behavior policy.

**Learning curves.** Learning curves of best tuned $\lambda$ for each dataset is presented in Figure 4 and Figure 5. Learning curves of different $\lambda$ for Gym-MuJoCo domains is presented in Figure 6.

Table 6: Hyperparameters of policy training in SPOT.

| | Hyperparameter | Value |
|---|---|---|
| | Optimizer | Adam [9] |
| | Critic learning rate | $3 \times 10^{-4}$ |
| | Actor learning rate | $3 \times 10^{-4}$ for Gym-MuJoCo |
| | | $1 \times 10^{-4}$ for AntMaze |
| TD3 | Batch size | 256 |
| | Discount factor | 0.99 |
| | Number of iterations | $10^6$ |
| | Target update rate $\tau$ | 0.005 |
| | Policy noise | 0.2 |
| | Policy noise clipping | 0.5 |
| | Policy update frequency | 2 |
| | Actor hidden dim | 256 |
| | Actor layers | 3 |
| Architecture | Actor dropout | 0.1 for Gym-MuJoCo |
| | | 0.0 for AntMaze |
| | Critic hidden dim | 256 |
| | Critic layers | 3 |
| SPOT | $\lambda$ | $\{0.05, 0.1, 0.2, 0.5, 1.0, 2.0\}$ Gym-MuJoCo |
| | | $\{0.025, 0.05, 0.1, 0.25, 0.5, 1.0\}$ AntMaze |

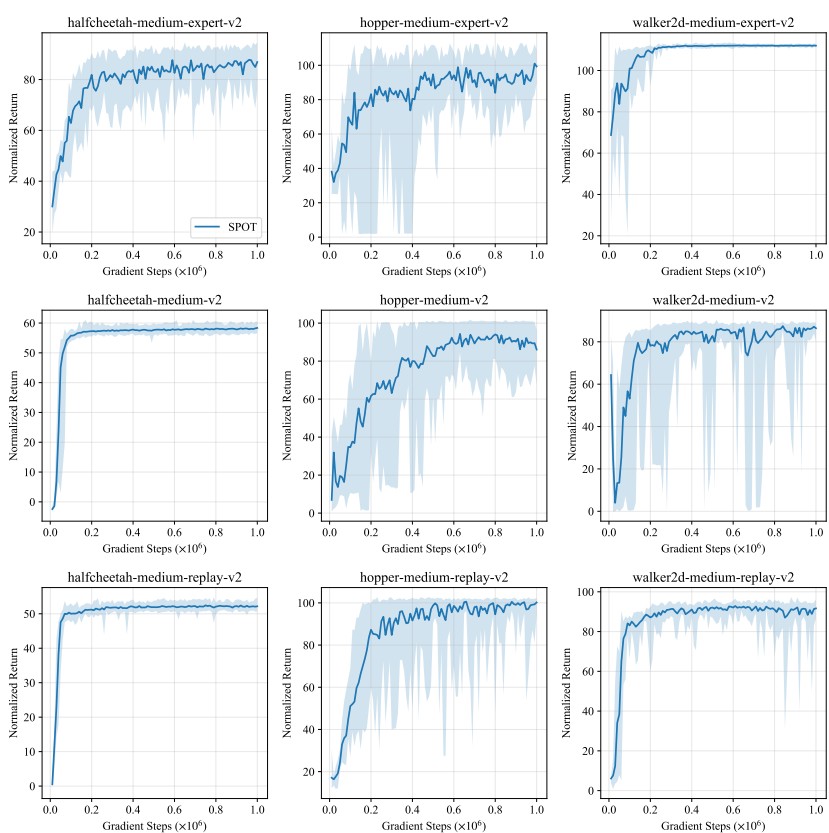

Figure 4: Learning curves of best-tuned $\lambda$ for Gym-MuJoCo domains. Error bars indicate min and max over 10 seeds.

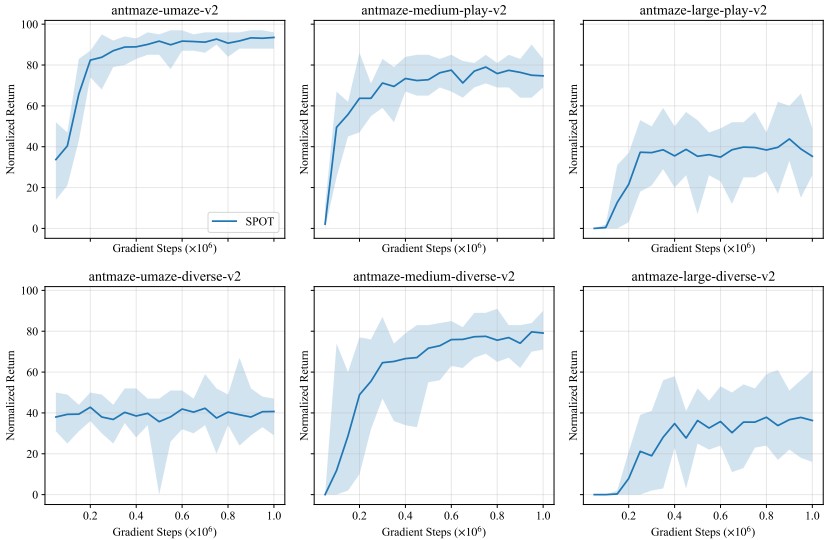

Figure 5: Learning curves of best-tuned $\lambda$ for AntMaze domains. Error bars indicate min and max over 10 seeds.

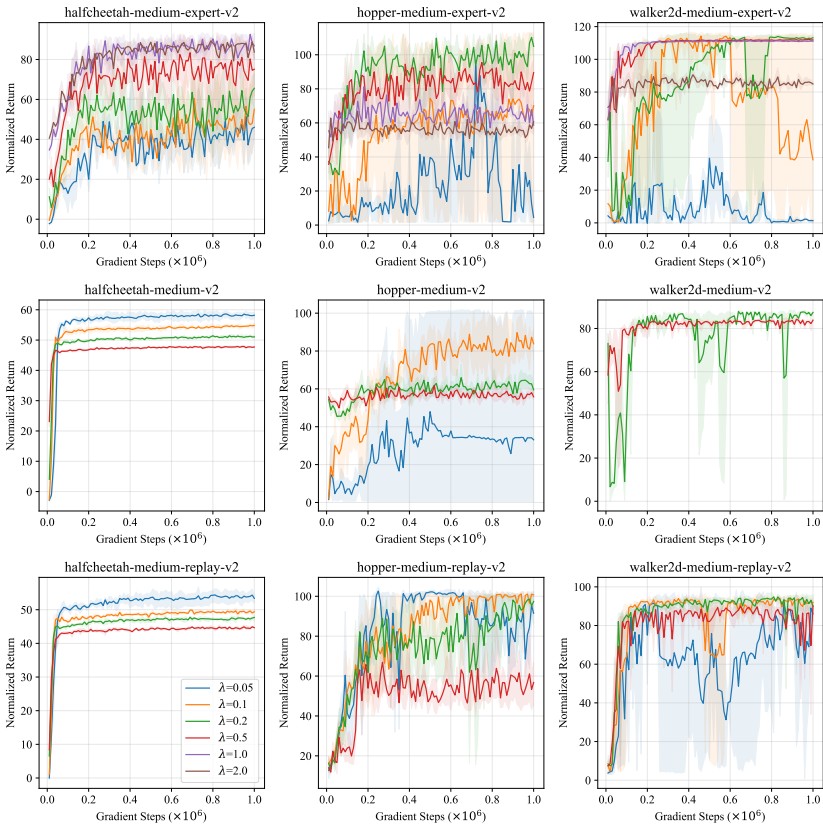

Figure 6: Learning curves of different $\lambda$ for Gym-MuJoCo domains. Error bars indicate min and max over 3 seeds. We only search for 4 values on "medium-replay" and "medium" datasets and SPOT fails to train on walker-medium-v2 dataset with $\lambda = 0.05$ and $\lambda = 0.1$ since the constraint is too loose.

## C.2 Ablation Study

**Ablation study with variants of SPOT (Figure 2).** We consider several variants of SPOT to perform an ablation study over the components in our method:

- 'Gaussian': We train a parameteric model $\widehat{\pi_\beta}$, which fits a tanh-Gaussian distribution to the actions $a$ given the states $s$: $\widehat{\pi_\beta}(\cdot|s) = \tanh \mathcal{N}(\mu(\cdot|s), \sigma(\cdot|s))$, following [11, 16]. Then we use $\widehat{\pi_\beta}$ as our density estimator in the actor learning objective. Hyperparameters for training Gaussian behavior models are almost the same as Table 5.

- 'w/o Q Norm': We remove the Q normalization trick from SPOT.

- 'Stoch w/ Ent': We adopt SAC [8] as the base off-policy method of SPOT. Training objectives can be written as:

$$J_Q(\theta) = \mathbb{E}_{(s,a,r,s')\sim\mathcal{D}} \left[ Q_\theta(s,a) - r - \gamma\mathbb{E}_{a'\sim\pi_\phi(s')}\left[ Q_{\bar{\theta}}(s',a')\right]\right]^2$$

$$J_\pi(\phi) = \mathbb{E}_{s\sim\mathcal{D}, a\sim\pi_\phi(s)} \left[ -Q_\theta(s,a) + \alpha \log \pi_\phi(a|s) - \lambda\widehat{\log \pi_\beta}(a|s; \varphi, \psi, L)\right].$$

The learning stochastic policy is parameterized as a tanh-Gaussian distribution and the temperature $\alpha$ is adjusted automatically, as [8]. We do not include the entropy term when performing the backup, as CQL [12]. Hyperparameters for training are almost the same as Table 6.

- 'Stoch w/o Ent': This variant is almost the same as the above one, except that we remove the entropy term:

$$J_\pi(\phi) = \mathbb{E}_{s\sim\mathcal{D}, a\sim\pi_\phi(s)} \left[ -Q_\theta(s,a) - \lambda\widehat{\log \pi_\beta}(a|s; \varphi, \psi, L)\right].$$

We present quantitative results of variants of SPOT in Table 7, corresponding to Figure 2.

Table 7: Quantitative results of ablation study on Gym-MuJoCo and AntMaze domains. For variants of SPOT, we report the mean and standard deviation for 5 seeds.

| | SPOT variants | | | | SPOT |
|---|---|---|---|---|---|
| | Gaussian | w/o Q Norm | Stoch w/o Ent | Stoch w/ Ent | |
| HalfCheetah-m-e-v2 | 77.7±8.4 | **87.8**±5.5 | 87.3±3.7 | 87.0±3.8 | 86.9±4.3 |
| Hopper-m-e-v2 | 79.9±21.7 | 98.6±16.4 | **103.1**±8.6 | 66.6±24.9 | 99.3±7.1 |
| Walker-m-e-v2 | 105.2±6.3 | **112.9**±0.9 | 111.7±0.8 | 110.9±0.4 | 112.0±0.5 |
| HalfCheetah-m-v2 | 57.2±0.6 | **63.0**±1.9 | 58.2±0.9 | 57.1±0.8 | 58.4±1.0 |
| Hopper-m-v2 | 96.6±3.2 | **101.2**±0.1 | 58.4±8.3 | 52.1±2.4 | 86.0±8.7 |
| Walker-m-v2 | 81.6±1.0 | 83.5±1.0 | 84.9±3.7 | 84.4±0.3 | **86.4**±2.7 |
| HalfCheetah-m-r-v2 | 53.3±1.4 | **53.9**±0.9 | 51.8±0.4 | 51.5±1.1 | 52.2±1.2 |
| Hopper-m-r-v2 | 95.2±10.4 | 98.5±6.1 | 82.7±12.8 | 99.3±3.7 | **100.2**±1.9 |
| Walker-m-r-v2 | 93.6±2.2 | **94.4**±2.6 | 84.7±6.4 | 86.1±10.3 | 91.6±2.8 |
| AntMaze-u-v2 | 83.0±4.9 | **95.2**±1.9 | 74.5±20.0 | 78.2±17.5 | 93.5±2.4 |
| AntMaze-u-d-v2 | 35.2±27.0 | 38.2±3.9 | 34.0±11.8 | 39.5±3.9 | **40.7**±5.1 |
| AntMaze-m-p-v2 | 46.0±13.8 | 72.2±5.3 | 20.0±28.6 | 21.0±36.4 | **74.7**±4.6 |
| AntMaze-m-d-v2 | 71.0±3.7 | 72.4±8.9 | 0.0±0.0 | 22.2±38.5 | **79.1**±5.6 |
| AntMaze-l-p-v2 | 29.6±4.8 | **38.8**±15.9 | 0.0±0.0 | 11.0±19.1 | 35.3±8.3 |
| AntMaze-l-d-v2 | 31.0±10.1 | 31.8±6.2 | 26.3±10.0 | 25.3±9.91 | **36.3**±13.7 |

**Effect of $L$ in density estimation.** As a way of investigating the effect of the tightness of density estimation (see Section 4.2) on the final performance, we evaluate different values of the number of samples $L$ used in density estimation. We present result of $L \in \{1, 5, 10\}$ for Gym-MuJoCo domains in Figure 7. We note that all variants yield similar performance, which demonstrates that for this circumstance the ELBO estimator is a good enough density estimator. Thus unless otherwise specified, we adopt $L = 1$ as a default option in all of our experiments.

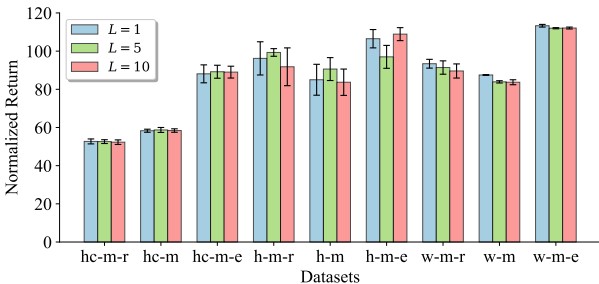

Figure 7: Comparing performance with different number of samples $L$ used in density estimation. hc = HalfCheetah, h = Hopper, w = Walker, m = medium, m-r = medium-replay, m-e = medium-expert. Bars indicate the mean and standard deviation for 3 seeds.

### C.3 Online Fine-tuning (Table 4)

**Baselines.** We takes the official implementation of IQL [10]: `https://github.com/ikostrikov/implicit_q_learning/` as our baseline.

**Implementation details.** We online fine-tune our models for 1M gradient steps after offline RL. In the online RL phase, we collect data actively in the environment with exploration noise $0.1$ and add the data to the replay buffer. We linearly decay the regularization term weight $\lambda$ in the online phase. We find that in the challenging AntMaze domains with high-dimensional state and action space as well as sparse reward, bootstrapping error is serious even in the online phase, thus we stop decay when $\lambda$ reaches the $20\%$ of its initial value at the 0.8M-th step. We also have experimented with gradually relaxing the implicit constraint of IQL by increasing its inverse temperature [13, 10] but we find that IQL does not benefit from this. Additionally, we find that a larger discount factor $\gamma$ is of great importance in the antmaze-large datasets, thus we set $\gamma = 0.995$ when fine-tuning on antmaze-large datasets, for both SPOT and IQL to ensure a fair comparison. All other training details are kept the same between the offline RL phase and the online RL phase.

**Learning curves.** Learning curves of online fine-tuning of SPOT and the baseline IQL are presented in Figure 8.

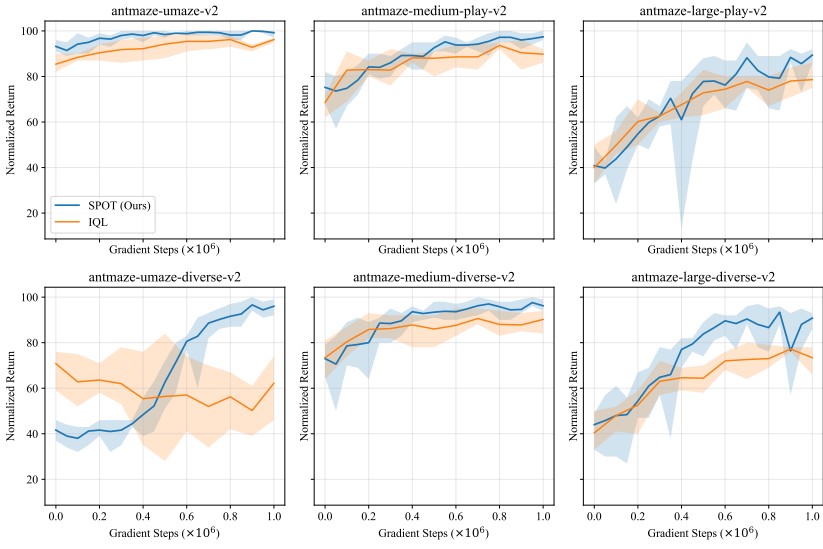

Figure 8: Learning curves of online fine-tuning for AntMaze domains of SPOT and the baseline IQL. Error bars indicate min and max over 5 seeds.

## C.4 Analytic Experiments (Figure 1b)

**Intuition.** We assume that the same constraint strength implies the same risk of extrapolation error on Q estimation (related theoretical bound can be found in [11]). Benefiting from the exact constraint formulation (Eq. (4)), SPOT can fully exploit feasible actions that is $\epsilon$-supported: $\{a : \pi_\beta(a|s) > \epsilon\}$. However, other kinds of constraints may deviate from the density-based formulation of $\epsilon$-support set, thus feasible actions under these constraints may only constitute a subset of the minimal support set that covers them. Under the risk of Q estimation error but only exploiting a subset of the $\epsilon$-supported actions, baseline methods may limit their optimality and provide a fragile tradeoff between satisfied constraint strength and optimality. To quantitatively illustrate this, we conduct experiments comparing the performance of different methods under the same constraint strength.

**Data.** We use the Gym-MuJoCo "medium-replay" and "medium" datasets from the D4RL benchmark [3], of the latest versions "v2".

**Baselines.** We choose several policy constraint methods as our baselines, taking their official implementation or the implementations evaluated by D4RL benchmarks:

- BCQ [6]: `https://github.com/rail-berkeley/d4rl_evaluations/tree/master/bcq`
- BEAR [11]: `https://github.com/rail-berkeley/d4rl_evaluations/tree/master/bear`
- PLAS [17]: `https://github.com/Wenxuan-Zhou/PLAS`
- TD3+BC [5]: `https://github.com/sfujim/TD3_BC`

**Constraint strength control.** Constraint strength of different policy constraint methods can be controlled by their own unique hyperparameters. By varying the values of these hyperparameters, we can get a spectrum of constraint strength. Hyperparameters that we adjust for each method are summaries as follows:

- BCQ [6]: max perturbation $\Phi \in \{0.02, 0.05, 0.1\}$. The smaller the value is, the stronger the constraint will be.
- BEAR [11]: MMD threshold $\varepsilon \in \{0.02, 0.05, 0.1\}$. The smaller the value is, the stronger the constraint will be.
- PLAS [17]: max latent action $\sigma \in \{1.0, 2.0, 3.0\}$. The smaller the value is, the stronger the constraint will be.
- TD3+BC [5]: Q term weight $\alpha \in \{1.0, 2.5, 4.0\}$. The smaller the value is, the stronger the constraint will be.
- SPOT (Ours): regularization term weight $\lambda \in \{0.05, 0.1, 0.2, 0.5\}$. The larger the value is, the stronger the constraint will be.

**Quantitative results.** We run baselines with varying hyperparameters for 3 seeds and present results in Table 8. We find that BEAR is unstable and rerunning for different seeds does not fix it, thus we exclude failed results from the Figure 1b and Figure 9.

Table 8: Quantitative results with 3 seeds for baselines in analytic experiments. hc = HalfCheetah, h = Hopper, w = Walker, m = medium, m-r = medium-replay.

| | BCQ | | | BEAR | | | PLAS | | |
|---|---|---|---|---|---|---|---|---|---|
| | $\Phi$=0.02 | $\Phi$=0.05 | $\Phi$=0.1 | $\varepsilon$=0.02 | $\varepsilon$=0.05 | $\varepsilon$=0.1 | $\sigma$=1.0 | $\sigma$=2.0 | $\sigma$=3.0 |
| hc-m-v2 | 44.2±1.6 | 46.7±0.4 | 49.3±0.8 | 42.8±0.1 | 42.8±0.1 | 17.3±28.8 | 43.3±0.3 | 44.8±0.1 | 45.0±1.0 |
| h-m-v2 | 55.2±2.1 | 59.5±1.4 | 54.3±2.7 | 51.0±1.4 | 51.9±2.8 | 1.9±1.9 | 56.9±7.4 | 55.0±3.2 | 52.9±5.5 |
| w-m-v2 | 80.9±1.4 | 70.3±11.1 | 69.2±1.1 | -0.2±0.1 | 19.8±34.7 | -0.3±0.0 | 74.1±2.3 | 78.4±4.9 | 73.4±8.3 |
| hc-m-r-v2 | 38.0±2.5 | 40.0±1.6 | 43.5±1.0 | 36.7±0.6 | 37.2±0.8 | 37.2±0.5 | 40.7±1.5 | 43.5±0.4 | 44.3±0.7 |
| h-m-r-v2 | 49.4±14.7 | 25.8±5.2 | 36.8±9.7 | 59.2±14.8 | 49.1±13.7 | 62.2±6.9 | 28.7±3.8 | 27.4±4.6 | 26.2±5.6 |
| w-m-r-v2 | 44.2±20.8 | 55.8±11.8 | 36.1±5.5 | 1.5±1.5 | 7.7±6.0 | 4.3±1.7 | 53.6±31.8 | 63.3±18.9 | 39.6±27.5 |

**Missing graphs.** Due to space limitation, we only present results for "medium-replay" datasets for Figure 1b. The complete graphs are presented in Figure 9.

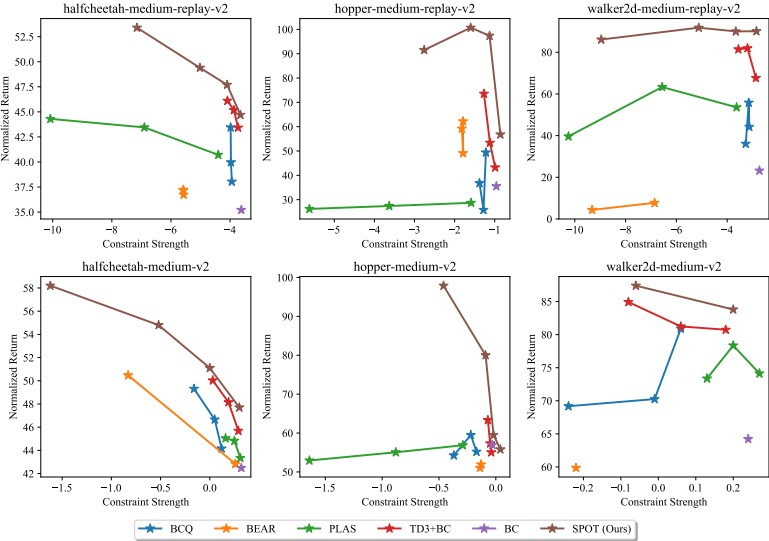

Figure 9: Full graphs of analysis on tradeoff between constraint strength and optimality, extending Figure 1b.

## C.5    Computation Cost

**Inferece time (Figure 3).** We evaluate the runtime of different offline RL methods, that interact with the HalfCheetah environment to produce a full 1000-steps trajectory. For parameterization methods, we evaluate BCQ (num. of sampled actions $N = 100$) [6], PLAS [17], EMaQ (num. of sampled actions $N = 100$) [7] and for regularization methods, we evaluate BEAR (num. of sampled actions $p = 10$) [11], TD3+BC [5] and our SPOT. All numbers of runtime of Figure 3 are the mean of 100 trajectories. We compare different methods with consistent model size to ensure fairness.

**Train time.** Table 9 presents train time of 1M steps of various offline RL algorithms. All train time experiments were run with author-provided implementations on a single TITAN V GPU and Intel Xeon Gold 6130 CPU at 2.10GHz.

Table 9: Train time of 1M steps of various offline RL algorithms.

|  | BCQ | BEAR | PLAS | CQL | TD3+BC | SPOT |
|---|---|---|---|---|---|---|
| Train time | 5h 25m | 12h 30m | 3h 5m | 14h 20m | 1h 58m | 3h 25m |

## D    Broader Impact

**Social impacts.** Offline reinforcement learning has the potential to enable or scale-up practical applications for reinforcement learning, such as robotics, recommendation, healthcare, or educational applications, where data collecting is always expensive or risky, and offline logged data can lead to a better real-world performance by either pure offline or offline2online learning. A limitation to offline RL is that the learned policy, regularized by the offline data, may contain biases originally from the data-collecting policy.

**Academic research.** Developing a simple and effective offline RL algorithm is the primary aim behind our work. We situate our work in the literature on policy constraint methods for offline RL, covering discussion w.r.t empirical performance, implementation simplicity, and computation efficiency. We identify that a standard off-policy RL algorithm plugged with a VAE-based explicit

support constraint is sufficient for exceeding most of substantially more complicated methods on both standard and challenging benchmarks, which may encourage researchers to revisit the progress of offline RL and derive new and better offline RL methods.