# OpenReview forum: "Supported Policy Optimization for Offline Reinforcement Learning"
_NeurIPS.cc/2022/Conference — NeurIPS 2022 Accept_

### Official Review · Reviewer_AVgj · 2022-06-29

**Rating:** 7
**Confidence:** 4
**Soundness:** 3 good
**Presentation:** 3 good
**Contribution:** 3 good

**Summary:**

The paper considers the offline Reinforcement Learning setting, and introduces a loss that allows any off-policy RL algorithm to learn from offline data. The loss addresses the issue of out-of-distribution actions being sampled by the actor being trained. The loss is obtained by first training a (conditional) Variational Autoencoder on state-action tuples. Then, its ELBO loss (used for training the VAE itself) is used as part of the actor training, to keep in close to the support of actions in the dataset. The critic training loss does not seem to have to be modified (Equation 1 is untouched), and leverages the actor to sample actions (so, indirectly, it samples actions in the support of the dataset).

An empirical evaluation in challenging environments shows that the proposed method, SPOT (the loss described above + TD3) outperforms all the baselines being considered. The results are sometimes moderate, sometimes much more significant.

**Questions:**

I don't have any question for the authors

**Limitations:**

There does not seem to be any negative societal impact of this paper that should be discussed. The proposed algorithm outperforms the baselines in every case, and is overall strong. As such, the lack of a "Limitations" section, for instance, is not problematic for this paper. A mention of possible areas of future improvement would have been nice in the conclusion, that for the moments sounds a bit like an advertisement.

**Strengths And Weaknesses:**

The proposed method is easy to implement, and indeed does not need to change fundamental aspects of an off-policy RL algorithm to make it offline.

Strengths (quality, clarity, somewhat the significance):

* The paper is well-written and easy to read. It flows naturally
* The proposed method is sound, and its derivation from constraint-based optimization is intuitive and motivates the resulting loss well.
* The empirical evaluation is thorough, with the impact of hyper-parameters being well-studied. The environments being used are challenging (not just toy environments), and many baselines are considered.
* Source code is provided! This is a rare event, and greatly helps reproducibility and answering questions such as "does the gradient flow to the actor through the VAE?" (the answer is yes). The code is also clean, self-contained and easy to understand.

There is no big weakness in this work, even though its novelty is maybe a bit lower than the other aspects listed in Strengths. While the proposed method is novel, and quite distinct from the existing literature, it is still using a VAE in combination with some off-policy RL algorithm. BCQ uses the VAE to sample actions (while this work uses the VAE as part of the actor loss). BRAC estimates the behavior policy and uses the KL divergence between the actor and the behavior policy to regularize learning. BEAR is inspired from constraint-based optimization like this work, but uses the maximum mean discrepancy to keep the actor close to the actions in the dataset. So, overall, the originality of this work, albeit satisfactory, cannot be stated as a strength.

Minor comments:

* Section 2 in the appendix is quite interesting, and if room can be made in the paper, would deserve to be in the main text
* The description of how $\lambda$ decreases over time in the online fine-tuning setting is interesting and should also be in the main text (it is in the appendix currently)

**Author response**

The authors mention that they will improve the conclusion of the paper and extend the main text, which were my two minor comments. The other reviewers also seem to be leaning towards acceptance, so I maintain my score of accept.

---

> ### Author Response · Authors · 2022-07-31
> **Response to Reviewer AVgj**
>
> We would like to sincerely thank Reviewer AVgj for providing the detailed review and the positive evaluation of the clarity, soundness and thoroughness of our paper.
>
> **Q1**: Rephrase the conclusion.
>
> We appreciate the nice suggestion about the conclusion. We have added discussion w.r.t. limitations and future work into the uploaded revision. Here is the revised conclusion:
>
> > 6 Conclusion
> >
> > We present Supported Policy OpTimization (SPOT), a policy constraint method to offline RL built upon off-the-shelf off-policy RL algorithms. Capturing the standard formulation of the support constraint, SPOT introduces a pluggable regularization term applied directly to the estimated behavior density and obtains excellent performance across different tasks in the D4RL benchmarks, including standard Gym-MuJoCo tasks and challenging AntMaze tasks. Furthermore, when online fine-tuned after offline RL pre-training, the pluggable design of our algorithm makes it seamless to take full advantage of well-established online methods and exceed the state-of-the-art on the challenging AntMaze domains. One limitation of our current method, shared by most policy constraint methods, is that the performance may be limited by the accuracy of estimation of the behavior policy. An exciting direction for future work would be to develop an effective pluggable constraint mechanism excluding explicit estimation of behavior policy.
>
> If the room can be made in the case of acceptance, we will enrich the conclusion with more possible areas of future improvement, for example, adaptive adjustment of constraint strength instead of manual tuning.
>
> **Q2**: Enrich the main text.
>
> We are glad that you highlight some parts of the supplementary material interesting. We will consider moving those contents into the main text if the final version permits an additional page, which is the tradition of NeurIPS.

---

### Official Review · Reviewer_qPyz · 2022-07-05

**Rating:** 6
**Confidence:** 2
**Soundness:** 3 good
**Presentation:** 3 good
**Contribution:** 3 good

**Summary:**

The paper present a regularization method for offline reinforcement learning called SPOT. The main idea is a policy constraint inspired by support constraint on behavior policy. The method is evaluated extensively on standard offline RL benchmarks and achieves state of the art result.

**Questions:**

A minor suggestion is on the symbols in $\log\pi_\beta(\pi_\phi(s)|s)$ (4), as $\pi_\beta$ is the probability $\in [0,1]$ while $\pi_\phi$ actually is the action $\in \mathcal{A}$. It may cause some confusion at first glance.

**Limitations:**

The authors have addressed the societal impact of this work.

**Strengths And Weaknesses:**

Strengths: The idea of support constraint looks simple and effective as in Sec.4.1, and the empirical result as shown in Figure 1 and Table 2 demonstrate its effectiveness for practical problems. From the empirical result, this work achieves new state-of-the-art results on standard offline RL benchmarks and is beneficial to the related communities.

Weakness: From my perspective, the effectiveness of the proposed algorithm is mainly demonstrated by empirical experiments. It would be more complete if the empirical improvement over previous methods can be justified with some theoretical analysis.

---

> ### Author Response · Authors · 2022-07-31
> **Response to Reviewer qPyz**
>
> Many thanks to Reviewer qPyz for providing the thorough review and valuable suggestions.
>
> **Q1**: Lack of theoretical analysis.
>
> The main concern raised by the reviewer is the lack of precise theoretical characterization of the empirical improvement. We agree that the paper is primarily methodological, which aims to propose a simple and pluggable offline RL method. SPOT brings a number of practical benefits for realistic applications, including simple implementation, inference efficiency and convenience for fine-tuning. Moreover, our empirical results are strong enough to be impactful. We would also like to point out that we do provide some preliminary effort to connect our method to the theoretical framework of support constraint in $\underline{\text{Section 3.2 and 4.1 of main paper}}$. While BEAR [1] uses a maximum mean discrepancy (MMD) constraint to approximate such support constraint, we empirically find that it is ineffective and our method SPOT faithfully and elegantly realizes the idea of support constraint in an explicit regularization based on behavior-density estimation, which is believed to contribute the performance boost.
>
> [1] Kumar et al. Stabilizing off-policy q-learning via bootstrapping error reduction. NeurIPS 2019.
>
> **Q2**: Comment on Equation 4.
>
> The notation $\log \pi_{\beta}\left(\pi_{\phi}(s) \mid s\right)$ is a bit confusing. As it is the most commonly used notation for deterministic and stochastic policies, we cannot think of a better notation. We have added a footnote to explain the current one in the uploaded revision. Thank you for pointing this out.

---

### Official Review · Reviewer_FcL9 · 2022-07-11

**Rating:** 6
**Confidence:** 5
**Soundness:** 3 good
**Presentation:** 3 good
**Contribution:** 3 good

**Summary:**

This paper proposes supported policy optimization (SPOT), which introduces a "pluggable" regularization term applied to an estimated behavior policy. In the implementation, following VAE, the authors use the ELBO to estimate the behavior policy, and following TD3+BC, add a normalization term to the policy loss. The authors emphasize that SPOT is computationally efficient at inference, obtains excellent performance, and enables effective online fine-tuning.

**Questions:**

1. The authors state that "our method benefits from a closer connection between theory and algorithm". What does the author mean by "theory"?
2. I think there is no essential difference between Equation 4 and Equation 3. Furthermore, the authors approximate Equation 4 and get Equation 5. Is the theoretical guarantee still satisfied?

**Limitations:**

yes

**Strengths And Weaknesses:**

Strengths:
+ This paper is well-written.
+ The proposed method is simple and intuitive.
+ The authors conduct extensive experiments to show the effectiveness of SPOT.

Weaknesses:
+ The paper lacks theoretical analysis.
+ The method is somewhat incremental.

---

> ### Author Response · Authors · 2022-07-31
> **Response to Reviewer FcL9**
>
> Many thanks to Reviewer FcL9 for providing the insightful review and valuable comments.
>
> **Q1**: Lack of theoretical analysis. What do we mean by "a closer connection between theory and algorithm"?
>
> While the paper is primarily methodological, our algorithmic designs are greatly inspired by relevant theoretical analysis. As argued by many previous works [1,2,3,4], support constraint may be sufficient to be theoretically and empirically effective for offline RL method. Theorem 4.1 in [1] shows that threshold $\epsilon$ of support set simultaneously trades off extrapolation error of Q estimation and performance of the constrained optimal policy. While BEAR [1] turns to using maximum mean discrepancy (MMD) to approximate such support constraint, we empirically find that it is ineffective. Our method SPOT directly converts the constraint $\pi_{\beta}\left(a | s\right)>\epsilon$ into a simple and straightforward regularization with strong empirical performance, which is believed to be meaningful in real applications.
>
> [1] Kumar et al. Stabilizing off-policy q-learning via bootstrapping error reduction. NeurIPS 2019.
>
> [2] Ghasemipour et al. EMaQ: Expected-max q-learning operator for simple yet effective offline and online rl. ICML 2021.
>
> [3] Laroche et al. Safe Policy Improvement with Baseline Bootstrapping. ICML 2019.
>
> [4] Liu et al. Provably Good Batch Reinforcement Learning Without Great Exploration. 2020.
>
> **Q2**: Derivation through Equations 3, 4, and 5.
>
> The constrained optimization problem (Equation 4) is exactly how we realize $\max_{a^{\prime}: \pi_{\beta}\left(a^{\prime} \mid s^{\prime}\right)>\epsilon} Q\left(s^{\prime}, a^{\prime}\right)$ in the definition of supported backup operator (Equation 3). As mentioned in $\underline{\text{Line 171-172 of main paper}}$, the approximation from Equation 4 into Equation 5 is heuristic, which does not satisfy a theoretical guarantee. However, we would like to highlight that it is commonly adopted by the derivation of previous work, including TRPO for online RL and BEAR, AWR for offline RL. Our empirical evaluation shows that this approximation is practically effective.
>
> **Q3**: On novelty.
>
> While the proposed algorithm is based on previous literature, including the theoretical framework of support constraint, well-established off-policy algorithms and a VAE-based density estimator, we conduct an elegant algorithmic design to achieve a practical and powerful method. We would also like to point out that our focus is meaningful and impactful. We do believe that a simple and pluggable design is sufficient to be effective for offline RL, eliminating implementation complexity, computational cost, or algorithmic gap with online RL, which is essential for practical applications.

---

> > ### Comment · Reviewer_FcL9 · 2022-08-08
> > **Reviewer response**
> >
> > Thank you for your response. The authors' response partially solved my questions, but I am still concerned about the novelty of the paper. I also notice that Reviewer AVgj has the same concern with this.

---

### Official Review · Reviewer_Eyz5 · 2022-07-11

**Rating:** 5
**Confidence:** 4
**Soundness:** 2 fair
**Presentation:** 3 good
**Contribution:** 2 fair

**Summary:**


The author presents Supported Policy OpTimization (SPOT), a policy constraint method to offline RL built upon an off-the-shelf off-policy RL algorithm. SPOT introduces a pluggable regularization term applied directly to the estimated behavior density. The experiments are conducted in the D4RL benchmark and several SOTA baselines are taken for comparison.


**Questions:**

1. In figure 1(b), why should we compare the performance of SPOT relative to the baselines under the same constraint strength? And why SPOT can achieve better performance with the same constraint strength since the related evidence cannot be found in the current theorem. btw, I think a meaningful comparison is that SPOT can easily adjust the degree of constraint by controlling \lambda, while other solutions do not do this well. Also, why are the other baselines only report a few points (e.g.,  In hopper-medium-replay-v2, PLAS only has one point)?
2. In figure 2, why does the effect become worse when using Gaussian, especially in Walker2d-medium and Hopper-medium. The policies of these datasets are Gaussian distribution, so Gaussian should be the most accurate modeling.
3. In the last paragraph of Section 5.2, you mentioned that the TD3 method is not as stable as SAC, especially in the Ant environment, and Figure 2 illustrates the problem. So, is it possible that the performance improvement of SPOT reported in Table 3 comes from our algorithm choice (TD3 instead of SAC) rather than our constraint mechanism? For a fair comparison, an ablation experiment may be required here.
4. In Section 5.3, the author compares the performance improvement of SPOT and IQL under the online fine-tuning setting, and the author claims that the SPOT algorithm can be online fine-tuned seamlessly. This point of view is a bit confusing to me: if I have an online reinforcement learning algorithm, then, for any offline reinforcement learning algorithm, I only need to pass the parameters of the optimal policy solved offline to the online reinforcement learning algorithm as the initialization parameters of the policy, and then I can achieve “fine-tuned seamlessness”. Is this something only SPOT can do? Further, I think the comparison with IQL is unfair. The author directly uses the constraint mechanism of IQL for online training, while the SPOT algorithm reduces the constraint coefficient. Since conservative updates are unnecessary during online training, this obviously suppresses the ability of IQL-based online-trained policies.
5. In Section 5.4, the authors compare the computation cost of inference of several algorithms and illustrate the inference speed advantage of SPOT. In my opinion, the advantage of SPOT's inference speed should exist in many algorithms, such as some algorithms based on Pluggable Regularization (such as CQL, BRAC, etc.) and most model-based offline reinforcement learning algorithms (such as MOPO). Is my statement correct? If so, I think the description in Section 5.4 is a bit overclaimed: we shouldn't consider it a SPOT-exclusive feature.



**Ethics Review Area:**

["I don’t know"]

**Limitations:**

I will consider increasing the score if the authors clarify the above questions or give a better experiment design to demonstrate the proposed method.

**Strengths And Weaknesses:**


Strengths:

1. the article is overall well-written and easy to follow.
2. The motivation of the article, simple and pluggable offline RL, is meaningful in real applications.
3. The solution is simple to implement and reasonable.


Weaknesses

The weaknesses come from the experiment designs, which are unclear to me. I have listed the related questions for the authors to qualify it.

---

> ### Author Response · Authors · 2022-07-31
> **Response to Reviewer Eyz5 (Part 1)**
>
> We would like to sincerely thank Reviewer Eyz5 for providing a detailed review and insightful questions.
>
> **Q1:** Why should SPOT be compared and achieve better performance under the same constraint strength?
>
> Here is the intuition behind the experiments: We assume that the same constraint strength implies the same risk of extrapolation error on Q estimation (related theoretical bound can be found in the BEAR paper). Benefiting from the exact constraint formulation ($\underline{\text{Eq. (4) of main paper}}$), SPOT can fully exploit feasible actions that is $\epsilon$-supported: $\\{a: \pi_{\beta}(a | s)>\epsilon\\}$. However, other kinds of constraints may deviate from the density-based formulation of $\epsilon$-support set, thus feasible actions under these constraints may only constitute a subset of the minimal support set that covers them. Under the risk of Q estimation error but only exploiting a subset of the $\epsilon$-supported actions, baseline methods limit their optimality and provide a fragile tradeoff between satisfied constraint strength and optimality. To quantitatively illustrate this, we plot Figure 1(b) and compare the performance of different methods under the same constraint strength.
>
> We have added the clarification of this intuition into Section 3.4 of the uploaded revision of supplementary material. All modifications are highlighted in blue.
>
> **Q2:** Why are the other baselines only report a few points in Figure 1(b)?
>
> First, for better illustration, we zoom in on the figure and therefore some outlier points are out of the figure (for example, PLAS on hopper-medium-replay). As mentioned in the caption of $\underline{\text{Figure 1 of main paper}}$, we present extended results in $\underline{\text{Figure 6 of  supplementary material}}$, which includes all points we reported.
>
> Second, as mentioned in $\underline{\text{Section 3.4 of supplementary material}}$, for some specific combinations of hyperparameter and algorithm (especially for BEAR), the constraint is too loose and the training diverges. We exclude these points from the figures.
>
> **Q3**: Why does the effect become worse when using Gaussian models on datasets generated by Gaussian policies?
>
> It is an interesting phenomenon, and we have tried our best to interpret it. Originally, we fit a tanh-Gaussian distribution with learned standard deviation to model behavior policies in our experiments of the main paper, since tanh-Gaussian is typically used for SAC to represent policies with bounded actions. However, we empirically find that if we alternatively use a standard (unbounded) Gaussian with fixed std (which is inspired by the design choice of our VAE decoder implementation), we result in $89.0\pm9.7$ on hopper-medium and $87.0\pm 1.6$ on walker-medium. These results are consistent with those of VAE. We suspect that tanh transformation over Gaussian introduces some optimization difficulties and a learned std may cause overfitting since each state-conditional action distribution has only one data point.
>
> Nevertheless, it is notable that **with either VAE-based or Gaussian density model**, SPOT substantially outperforms state-of-the-art offline RL baselines on Gym-MuJoCo medium and medium-replay datasets ($\underline{\text{Table 2 of main paper}}$ and $\underline{\text{Table 3 of supplementary material}}$), which demonstrates the effectiveness of our proposed explicit density-based regularization. Besides, VAE has the advantage of modeling more complex distributions.

---

> > ### Author Response · Authors · 2022-07-31
> > **Response to Reviewer Eyz5 (Part 2)**
> >
> > **Q4**: How does our constraint mechanism contribute to performance improvement?
> >
> > We have compared with baseline methods built upon TD3, including BCQ, PLAS, TD3+BC on Gym-MuJoCo datasets and BCQ, TD3+BC on AntMaze datasets in $\underline{\text{Figure 1, Tables 2 and 3 of main paper}}$. We further evaluate PLAS on AntMaze datasets. Here are the results, which are also included in the uploaded revision:
> >
> > |      | Am-u               | Am-ud              | Am-mp              | Am-md              | Am-lp              | Am-ld               | Total                |
> > | ---- | ------------------ | ------------------ | ------------------ | ------------------ | ------------------ | ------------------- | -------------------- |
> > | PLAS | 62.0 $\pm$ 16.7    | **45.4** $\pm$ 7.9 | 31.4 $\pm$ 21.5    | 20.6 $\pm$ 27.7    | 2.2 $\pm$ 3.8      | 3.0 $\pm$ 6.7       | 164.6 $\pm$ 84.3     |
> > | SPOT | **93.5** $\pm$ 2.4 | 40.7 $\pm$ 5.1     | **74.7** $\pm$ 4.6 | **79.1** $\pm$ 5.6 | **35.3** $\pm$ 8.3 | **36.3** $\pm$ 13.7 | **359.6** $\pm$ 39.7 |
> >
> > *Note: Am = AntMaze, u = umaze, m = medium, l = large, d = diverse, p = play. SPOT results are from the main paper.*
> >
> > SPOT significantly outperforms these methods on both domains. Notably, all baseline methods based on TD3 provide poor performance on the AntMaze domain (total score 142.8 for BCQ, 120.2 for TD3+BC, 164.6 for PLAS). These results illustrate that the proposed constraint mechanism, namely direct regularization on behavior density of learned actions, essentially contributes to SPOT's performance on complex offline RL tasks.
> >
> > Furthermore, the proposed constraint mechanism combined with SAC works comparably to the original TD3 variant on Gym-MuJoCo. It outperforms baseline methods as well, which also shows the effectiveness of the proposed constraint mechanism.
> >
> > **Q5:** Advantage of SPOT for online fine-tuning "seamlessly."
> >
> > Here by "seamlessly," we mean to fully exploit well-established powerful online RL algorithms when online fine-tuned, with a minimal algorithmic gap, eliminating unnecessary complexity or hyperparameter tuning. We argue that not all offline RL methods can realize this idea well.
> >
> > (1) Policy constraint methods via parameterization, such as BCQ, introduce additional structure into the policy. Thus standard online RL algorithms that typically formulate the policy as a simple fully-connected network, cannot be initialized directly by these offline trained policies.
> >
> > (2) Even if we can initialize with the offline pretrained policy, [1] indicates that off-policy bootstrapping error can cause an initial performance decrease when online fine-tuned with a standard off-policy method. Thus we also need a conservative constraint, which is typically a component of offline RL methods.
> >
> > (3) However, If we directly use an offline RL algorithm for online fine-tuning, performance or training speed may be limited by excessive conservatism (see the IQL experiments below, for example). Further, how these offline methods perform when online fine-tuned is not fully understood, tuned, and benchmarked by the community.
> >
> > Our method SPOT is built upon standard off-policy algorithms (with well-established implementation and hyperparameters), and a single hyperparameter $\lambda$ can easily control the strength of the pluggable constraint. Both advantages, as well as strong offline performance, contribute to the superior online performance of SPOT.
> >
> > [1] Ashvin Nair *et al*. AWAC: Accelerating online reinforcement learning with offline datasets.

---

> > > ### Author Response · Authors · 2022-07-31
> > > **Response to Reviewer Eyz5 (Part 3)**
> > >
> > > **Q6:** Comparison with IQL under the online fine-tuning setting.
> > >
> > > We agree with the reviewer that the conservatism in IQL may limit its online performance. IQL learns policy using advantage-weighted regression [1], whose implicit KL-divergence constraint is controlled by inverse-temperature $\beta$  in IQL. Thus, like how we gradually relax the constraint of SPOT, we further evaluate a variant of IQL whose inverse-temperature $\beta$ is increased linearly to $2\beta$  during online fine-tuning. Here are the results:
> > >
> > > |                   | IQL w/ constraint relax       | SPOT (from main paper)                  |
> > > | ----------------- | ----------------------------- | --------------------------------------- |
> > > | Umaze-v2          | 86.8 $\rightarrow$ 96.6       | **93.2** $\rightarrow$ **99.2** (+2.6)  |
> > > | Umaze-diverse-v2  | **68.0** $\rightarrow$ 42.8   | 41.6 $\rightarrow$ **96.0** (+53.2)     |
> > > | Medium-play-v2    | 73.4 $\rightarrow$ 92.8       | **75.2** $\rightarrow$ **97.4** (+4.6)  |
> > > | Medium-diverse-v2 | **73.8** $\rightarrow$ 90.4   | 73.0 $\rightarrow$ **96.2** (+5.8)      |
> > > | Large-play-v2     | **43.2** $\rightarrow$ 77.6   | 40.8 $\rightarrow$ **89.4** (+11.8)     |
> > > | Large-diverse-v2  | 41.4 $\rightarrow$ 78.8       | **44.0** $\rightarrow$ **90.8** (+12.0) |
> > > | **Total**         | **386.6** $\rightarrow$ 479.0 | 367.8 → **569.0** (+90.0)               |
> > >
> > > As shown in the above table, constraint relaxation does not help IQL on online fine-tuning, and SPOT still shows superior performance. We have added comments about this result in Section 3.3 of the uploaded revision of supplementary material.
> > >
> > > As mentioned above, one advantage of SPOT's pluggable design is that we can totally remove the constraint if necessary (for example, when the bootstrap error is not severe) and restore a standard off-policy algorithm. However, implicit constraint serves as a native component in IQL. With $\beta$ approaching infinity, there exist extreme numerical issues with exponential weighting. While this benifit is shared by all methods based on pluggable regularization, SPOT has the strongest offline performance.
> > >
> > > [1] Xue Bin Peng *et al*. Advantage-weighted regression: Simple and scalable off-policy reinforcement learning.
> > >
> > > **Q7:** Computation cost of inference.
> > >
> > > We agree with the reviewer that many methods also only need one forward pass of the policy network to do inference, thus making inference efficient. In $\underline{\text{Section 5.4 of main paper}}$, we aim to highlight the comparison with policy constraint methods via parameterization (BCQ, EMaQ, PLAS) that also utilize explicit density constraint but couple the policy with generative models or critic network ($\underline{\text{Section 2 of supplementary material}}$). SPOT enjoys the best of both worlds: explicit density-based constraint and inference efficiency. We have clarified this claim in the uploaded revision.

---

> > ### Comment · Reviewer_Eyz5 · 2022-08-07
> > **Response**
> >
> > Thanks for your elaborate response and the added experiments. The response solves most of my concerns. I have further questions on Q5 and Q6.  Could you show the hyper-parameter you tuned on IQL for constraint relaxation and the corresponding results? Besides, I am curious about the results of IQL on the online fine-tuning setting by removing the relaxation.

---

> > > ### Author Response · Authors · 2022-08-08
> > > **Response to Reviewer Eyz5**
> > >
> > > Thank you for your interactive response, which gives us further opportunity to answer your questions!
> > >
> > > For IQL with constraint relaxation on the online fine-tuning setting, we took the official implementation and hyperparameters of IQL except that we tried to linearly increase the inv-temperature $\beta$ of IQL during online fine-tuning to relax the implicit constraint, which enables a fair comparison with SPOT. We have experimented with linearly increasing the original $\beta=10.0$ to $2\beta$ ($=20.0$), $3\beta$ ($=30.0$) and $5\beta$ ($=50.0$), during 1M online steps. Besides, for IQL without constraint relaxation, this is exactly the experimental setting of the original IQL with a fixed $\beta$ and we have presented the corresponding results in $\underline{\text{Table 4 of main paper}}$. We summarize here all experimental results, including initial performance after offline RL and performance after 1M online steps:
> > >
> > > |                   | IQL (fixed $\beta$)                   | IQL ($\beta \rightarrow 2\beta$)   | IQL ($\beta \rightarrow 3\beta$)               | IQL ($\beta \rightarrow 5\beta$)      | SPOT                                        |
> > > | ----------------- | ------------------------------------- | ---------------------------------- | ---------------------------------------------- | ------------------------------------- | ------------------------------------------- |
> > > | Umaze-v2          | 85.4 $\rightarrow$ 96.2               | 86.8 $\rightarrow$ 96.6            | 87.3 $\rightarrow$ 97.0                        | 89.3 $\rightarrow$ $\underline{98.0}$ | 93.2 $\rightarrow$ $\textbf{99.2}$          |
> > > | Umaze-diverse-v2  | 70.8 $\rightarrow$ 62.2               | 68.0 $\rightarrow$ 42.8            | 74.0 $\rightarrow$ 63.3                        | 69.0 $\rightarrow$ $\underline{64.0}$ | 41.6 $\rightarrow$ $\textbf{96.0}$          |
> > > | Medium-play-v2    | 68.6 $\rightarrow$ 89.8               | 73.4 $\rightarrow$ 92.8            | 68.0 $\rightarrow$ $\underline{93.7}$          | 68.7 $\rightarrow$ 90.3               | 75.2 $\rightarrow$ $\textbf{97.4}$          |
> > > | Medium-diverse-v2 | 73.4 $\rightarrow$ 90.2               | 73.8 $\rightarrow$ 90.4            | 63.0 $\rightarrow$ $\underline{94.0}$          | 71.0 $\rightarrow$ 92.0               | 73.0 $\rightarrow$ $\textbf{96.2}$          |
> > > | Large-play-v2     | 40.0 $\rightarrow$ $\underline{78.6}$ | 43.2 $\rightarrow$ 77.6            | 46.3 $\rightarrow$ 75.0                        | 38.3 $\rightarrow$ 75.3               | 40.8 $\rightarrow$ $\textbf{89.4}$          |
> > > | Large-diverse-v2  | 40.4 $\rightarrow$ 73.4               | 41.4 $\rightarrow$ 78.8            | 50.0 $\rightarrow$ $\underline{84.3}$          | 44.0 $\rightarrow$ 78.0               | 44.0 $\rightarrow$ $\textbf{90.8}$          |
> > > | **Total**         | 378.6 $\rightarrow$ 490.4$\pm$25.8    | 386.6 $\rightarrow$ 479.0$\pm$41.2 | 388.7 $\rightarrow$ $\underline{507.3}\pm$22.2 | 380.3 $\rightarrow$ 497.7$\pm$27.0    | 367.8 $\rightarrow$ $\textbf{569.0}\pm$12.4 |
> > >
> > > We would also like to highlight that the constraint relaxation schedule of SPOT ($\lambda$ linearly decayed to $0.2 \lambda$) is set as common practices without careful tuning. A careful tuning of this may provide even stronger fine-tuning performance of SPOT.
> > >
> > > Please let us know what else we can do to address any lingering issues. We'd be happy to answer your future questions.

---

> ### Author Response · Authors · 2022-08-09
> **Discussion period ends soon**
>
> Dear Reviewer,
>
> It is a kind reminder that **this is the last day of the one-week Reviewer-author discussion**. Following your suggestion, we believe that we have made a great effort to provide all the experiments and clarifications that we can.
>
> If you have read our **latest response**, please kindly let us know. Any further questions/discussions are welcome.
>
> Thanks again for your review. Looking forward to your reply. Thank you!

---

> > ### Comment · Reviewer_Eyz5 · 2022-08-09
> > **Response**
> >
> > Thanks for the detailed response. Most of my concerns are solved. I will increase my score.

---

> > > ### Author Response · Authors · 2022-08-09
> > > **Thanks for the Response of Reviewer Eyz5**
> > >
> > > We'd like to thank you again for your time and efforts in providing a valuable review and carefully judging our feedback. We really enjoy the communication, and it helps us make our paper better.

---

### Author Response · Authors · 2022-08-01
**Summary of Revision**

We would like to thank the reviewers for their detailed and insightful comments. In this paper, we aim to propose a simple and pluggable offline RL method with a number of practical benefits for realistic applications, including implementation simplicity, strong empirical performance, and convenience for online fine-tuning.

We have made every effort to address all the reviewers' concerns and responded to the individual reviews below. We have also updated the paper with several modifications to address reviewer suggestions and concerns. Summary of updates:

1. We added PLAS results to Table 3;
2. We enriched the conclusion (Section 6) with discussion w.r.t. limitations and future work;
3. We expanded the discussion in Section 5.4 to clarify our claim about inference efficiency;
4. We added a footnote to explain our notation $\log \pi_{\beta}\left(\pi_{\phi}(s) | s\right)$ in Equation 4;
5. We added clarification of the intuition behind Figure 1(b) in Section 3.4 of supplementary material;
6. We added comments that constraint relaxation empirically does not help IQL for fine-tuning in Section 3.3 of supplementary material.

All updates are highlighted in blue.

---

### Meta-Review · Area_Chair_vXZ4 · 2022-08-25

**Recommendation:** Accept
**Confidence:** Certain

**Metareview:**

This work presents an interesting idea of constraining the policy network in offline reinforcement learning (RL)  to not only be within the support set but also avoid the out-of-distribution actions effectively unlike the standard behavior policy through behavior regularization.
The proposed Supported Policy OpTimization (SPOT) method leverages the theoretical framework of density-based support constraint. and adopts a VAE-based density estimator to model the supports of behavioral actions. Such a simple method indeed allows effective density-based regularization and can be flexibly be combined with most standard off-policy RL algorithms. Experiments also show that the propose algorithm achieves better performances than SOTA offline RL methods. All the reviewers think that the paper is written carefully, with the ideas explained intuitively, and algorithms tested extensively to showcase the effectiveness of SPOT. Therefore the consensus is to accept this paper for publication at NeurIPS22.


**Award:**

No

---

### Decision · Program_Chairs · 2022-09-14

Accept